# Rapid odor processing by layer 2 subcircuits in lateral entorhinal cortex

Sebastian H Bitzenhofer*[†], Elena A Westeinde, Han-Xiong Bear Zhang, Jeffry S Isaacson*

Center for Neural Circuits and Behavior and Department of Neurosciences, University of California, San Diego, La Jolla, United States

**Abstract** Olfactory information is encoded in lateral entorhinal cortex (LEC) by two classes of layer 2 (L2) principal neurons: fan and pyramidal cells. However, the functional properties of L2 cells and how they contribute to odor coding are unclear. Here, we show in awake mice that L2 cells respond to odors early during single sniffs and that LEC is essential for rapid discrimination of both odor identity and intensity. Population analyses of L2 ensembles reveal that rate coding distinguishes odor identity, but firing rates are only weakly concentration dependent and changes in spike timing can represent odor intensity. L2 principal cells differ in afferent olfactory input and connectivity with inhibitory circuits and the relative timing of pyramidal and fan cell spikes provides a temporal code for odor intensity. Downstream, intensity is encoded purely by spike timing in hippocampal CA1. Together, these results reveal the unique processing of odor information by LEC subcircuits and highlight the importance of temporal coding in higher olfactory areas.

*For correspondence:
sebastian.bitzenhofer@zmnh.uni-hamburg.de (SHB);
jisaacson@ucsd.edu (JSI)

Present address: [†]Institute of Developmental Neurophysiology, Center for Molecular Neurobiology, University Medical Center Hamburg-Eppendorf, Hamburg, Germany

Competing interest: The authors declare that no competing interests exist.

## Editor's evaluation

This work provides rigorous and high quality data regarding how neurons in the lateral entorhinal cortex (LEC) represent odor information as well as supporting a critical role of this area in odor discrimination. The LEC is an understudied area of the brain, yet critical for odor associations. This work will be of great interest to a wide neuroscience audience.

## Introduction

Olfactory cues provide rich information about the environment critical for behaviors as diverse as food seeking, social interactions, and predator avoidance. These behaviors depend not only on the identification of specific odors, but also on the detection of odor concentration which is essential for odor-guided navigation (*Ache et al., 2016*; *Marin et al., 2021*). In mammals, olfactory information is initially encoded by the firing activity of olfactory bulb (OB) mitral cells (*Uchida et al., 2014*; *Wilson and Mainen, 2006*). This information projects directly via the lateral olfactory tract (LOT) to the primary olfactory (piriform) cortex (PCx), a region thought to be critical for odor perception (*Blazing and Franks, 2020*; *Uchida et al., 2014*; *Wilson and Sullivan, 2011*). Although the properties of circuits in OB and PCx governing olfaction have received considerable attention, much less is known about odor processing in higher brain areas. In this study, we explore the nature of odor coding in lateral entorhinal cortex (LEC), a higher region that transmits information to the hippocampus which underlies odor-dependent memories and navigation (*Li et al., 2017*; *Radvansky and Dombeck, 2018*).

The LEC receives two main sources of olfactory input: direct projections from OB mitral cells via the LOT and indirect projections via PCx principal cells (*Haberly and Price, 1978*; *Kerr et al., 2007*). These inputs synapse onto two distinct types of layer 2 (L2) principal neurons: fan and pyramidal cells (*Canto and Witter, 2012*; *Kobro-Flatmoen and Witter, 2019*; *Tahvildari and Alonso, 2005*). Fan

cells in L2a have extensive apical dendritic arbors but lack basal dendrites, express reelin, and project via the lateral perforant path to granule cells in the dentate gyrus (*Leitner et al., 2016*; *Vandrey et al., 2020*). Pyramidal cells are concentrated in L2b, express calbindin, project to stratum lacunosum moleculare of hippocampal CA1, and send feedback projections to the PCx and OB (*Chapuis et al., 2013*; *Leitner et al., 2016*). LEC neurons respond in an odor-specific fashion (*Leitner et al., 2016*; *Woods et al., 2020*; *Xu and Wilson, 2012*) and calcium imaging in anesthetized mice revealed that fan cells respond more selectively to odors than pyramidal cells (*Leitner et al., 2016*). However, the role of LEC in odor discrimination has been debated (*Chapuis et al., 2013*; *Lee et al., 2021*; *Li et al., 2017*) and the coding of odor identity and intensity by fan and pyramidal cells in awake animals has not been established.

Rodents can make odor-guided behavioral decisions based on neural activity triggered by a single sniff (*Chong and Rinberg, 2018*; *Uchida and Mainen, 2003*) and studies have shed light on odor coding features of the OB and PCx. Odor identity is represented by the spatiotemporal activity patterns of OB mitral cells (*Bathellier et al., 2008*; *Chong and Rinberg, 2018*; *Cury and Uchida, 2010*; *Shusterman et al., 2011*) and the spike timing of cells early after odor inhalation is especially important for odor discrimination (*Chong et al., 2020*; *Gill et al., 2020*; *Wilson et al., 2017*). Changes in odor concentration can be decoded from both spike rate and the temporal profile of odor-evoked responses (*Bathellier et al., 2008*; *Cang and Isaacson, 2003*; *Margrie and Schaefer, 2003*; *Sirotin et al., 2015*). In PCx, odor identity is represented by distributed ensembles of active layer 2/3 pyramidal cells (*Blazing and Franks, 2020*; *Miura et al., 2012*; *Roland et al., 2017*; *Schoonover et al., 2021*; *Stern et al., 2018*; *Stettler and Axel, 2009*; *Uchida et al., 2014*). Calcium imaging in anesthetized mice revealed that while PCx population response patterns become markedly different over a 100-fold change in odor concentration, responses in a subset of neurons are insensitive to concentration (*Roland et al., 2017*). PCx recordings in awake mice indicate that odor identity is encoded by ensembles of pyramidal cells active early during individual sniffs in a manner invariant to odor concentration (*Bolding and Franks, 2018*). In contrast, odor intensity could be determined from a subpopulation of later responding pyramidal cells whose firing latencies shift earlier as odor concentration increases (*Bolding and Franks, 2017*).

In this study, we use extracellular recordings in awake mice, odor-driven behavior, and brain slice experiments to determine how LEC L2 principal cells contribute to olfactory information processing.

## Results

### Rapid odor-evoked activity during single sniffs in LEC

Previous work revealed distinct differences in the timing of odor-evoked activity in the OB and PCx during individual sniffs. While OB mitral cells fire spikes at odor-specific latencies that tile individual respiration cycles (*Bathellier et al., 2008*; *Cury and Uchida, 2010*; *Shusterman et al., 2011*), PCx pyramidal cells preferentially fire within the first 100 ms of odor inhalation (*Bolding and Franks, 2018*). However, the timing of odor-evoked activity in LEC relative to respiration-coupled firing in the OB and PCx is unknown. To address this question, we made acute simultaneous recordings of multiunit activity using silicon probes in the OB mitral cell layer, L2/3 of anterior PCx, and L2 of LEC in awake, head-fixed mice (*n* = 6 recordings from five mice, *Figure 1a*, *Figure 1—figure supplement 1*). We monitored nasal airflow at the outlet of a custom built olfactometer and delivered monomolecular odors (*n* = 11) triggered by the exhalation phase of respiration (respiration cycle duration ~300 ms, *Figure 1—figure supplement 1*). This approach allowed us to precisely align all recordings to the onset of the first inhalation of the applied odor for each trial (*Figure 1b*). Consistent with recent findings (*Bolding and Franks, 2018*), while odors evoked OB mitral cell activity during both early and late phases of a sniff (peaks at 43 and 232 ms, respectively), responses in PCx selectively occurred early (peak time 53 ms) after odor inhalation (*Figure 1c*). Interestingly, we found that simultaneously recorded activity in LEC also preferentially occurred early after odor inhalation (peak at 75 ms, *Figure 1c*). Experiments using channelrhodopsin-driven firing of OB mitral cells revealed similar shifts in the timing of early activity from OB to LEC suggesting that increasing latencies largely reflect axonal conduction times across regions (*Figure 1—figure supplement 1*).

We isolated single units to further explore the timing of odor-evoked activity across individual cells in the three brain regions (*Figure 1d–f*). While individual OB mitral cells showed odor-evoked

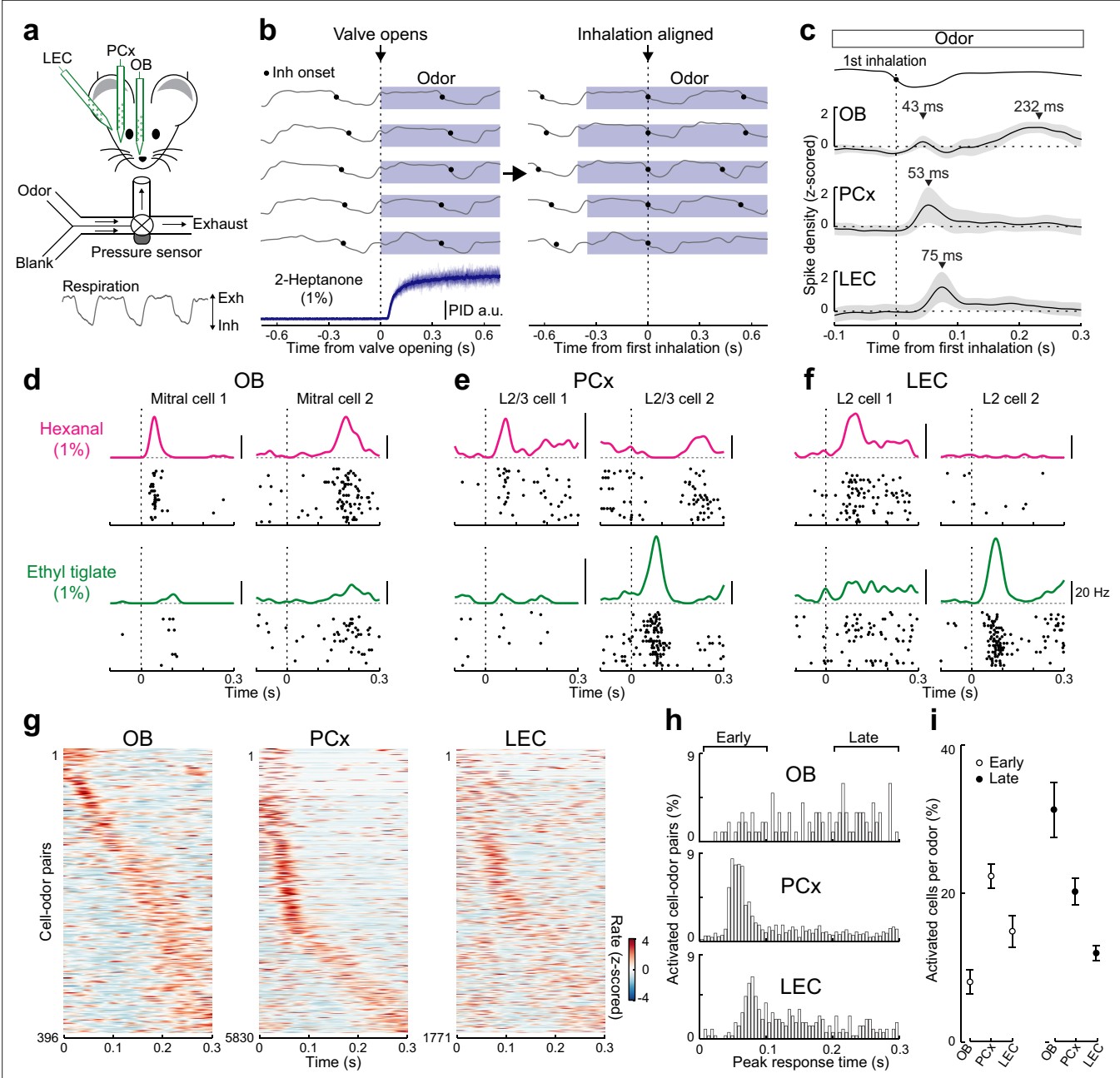

**Figure 1.** Rapid responses dominate odor-evoked activity in lateral entorhinal cortex (LEC) layer 2 (L2). (**a**) Experimental setup for triple recordings of odor-evoked activity in the olfactory bulb (OB), anterior PCx, and LEC of awake head-fixed mice. Downward and upward deflections on respiration trace from pressure sensor indicate inhalation (Inh) and exhalation (Exh), respectively. (**b**) Left, respiration signals during odor delivery (blue shading) for five trials are plotted above PID measurement of odor (2-Heptanone) at the sampling port. Black circles, inhalation onset. Right, same respiration signals aligned to onset of first inhalation. (**c**) OB mitral cell odor responses during individual sniffs have both early and late components, while early responses dominate in PCx and LEC. Spike-density plots of single-unit activity averaged across all triple recordings (n = 6 recordings from five mice) from OB (n = 36 cells), PCx (n = 530 cells), and LEC (n = 161 cells) responses to 11 odors. Trials were aligned to the first inhalation during each odor (dashed line). Top trace, representative averaged respiration trace from one experiment. (**d–f**) Raster plots and spike-density functions of odor-evoked activity for two representative cells from OB, PCx, and LEC in response to hexanal (top) and ethyl tiglate (bottom). (**g–i**) OB mitral cell responses occur evenly across the entire respiration cycle, while responses of PCx L2/3 cells and LEC L2 cells often occur early following odor inhalation. Plots reflect odor-evoked firing rates for all cell–odor pairs rank ordered by time of peak firing. For cross-validation, peristimulus time histograms are plotted from even trials that are sorted relative to the peaks obtained from odd trials. (**g**) Plots of odor-evoked firing rates for all cell–odor pairs rank ordered by time of peak firing for OB mitral cells (left), PCx L2/3 cells (middle), and LEC L2 cells (right). (**h**) Histograms showing time of peak odor responses relative to inhalation onset for significantly activated cell–odor pairs in the OB, PCx, and LEC. (**i**) Percent of activated cells per odor in the OB, PCx, and LEC during the early (0–100 ms)

*Figure 1 continued on next page*

*Figure 1 continued*

and late (200–300 ms) phase of odor-evoked activity marked in (**h**). Error bars represent standard error of the mean (SEM).

The online version of this article includes the following source data and figure supplement(s) for figure 1:

**Source data 1.** Source data for *Figure 1*.

**Figure supplement 1.** Recording sites, respiration, and propagation of optogenetic activation of mitral cells.

**Figure supplement 1—source data 1.** Source data for *Figure 1—figure supplement 1*.

responses that peaked at times that tiled the entire respiration cycle, many L2/3 PCx and L2 LEC cells preferentially responded early after odor inhalation (*Figure 1g–h*). Overall, the percentage of significantly activated cells per odor during the early phase of respiration (<100 ms after inhalation) was greater in the PCx and LEC compared to the OB (*Figure 1i*). In contrast, across the three olfactory areas, the fraction of activated cells per odor during later times (200- to 300-ms postinhalation) was lowest for the LEC. Together, these results suggest that activity triggered early during individual sniffs is important for odor coding in LEC.

## Behavioral discrimination of odor identity and intensity requires the LEC

To test the importance of the LEC in odor-driven behavior, we took advantage of an optogenetic approach to acutely silence the LEC in head-fixed mice performing odor discrimination in a two-alternative forced-choice (2AFC) task (*Figure 2a*, *Figure 2—figure supplement 1*). We used transgenic mice (*Gad2^{cre}* × Ai32) selectively expressing ChR2 in inhibitory neurons and made bilateral optical windows over the LEC for photostimulation (Materials and methods). Mice were trained (>75% correct responses) to lick left or right to distinguish odor identity (isoamyl acetate vs. limonene, both at 1% concentration). For each trial (≥4-s interval), odor application (1-s duration) began at the respiration exhalation phase so that behavior could be aligned to the onset of the first inhalation of each odor. Mice were free to report their choice (left or right lick) at any time within 2 s of odor onset and fiber-coupled LEDs (473 nm) targeting the LEC bilaterally were used to suppress cortical activity during odor delivery on a random subset (25%) of trials. We found that LEC is critical for discrimination of odor identity since LEC silencing reduced the fraction of correct responses to the chance level (*Figure 2b*, LED off: 83.76 ± 3.17%, LED on: 57.06 ± 5.22%, p = 0.009, paired *t*-test, *n* = 6 mice). Moreover, by analyzing lick timing, we determined that the discriminability measure *d'* became significantly different under control conditions as early as 225 ms after inhalation onset and performance accuracy increased within 200 ms of inhalation (*Figure 2b*, *Figure 2—figure supplement 1*). This rapid behavioral discriminability is consistent with the rapid timing of odor-evoked activity we observe in LEC, especially when considering motor initiation delays.

We next tested whether LEC played a role in the behavioral discrimination of odor intensity. Mice performed the same 2AFC task, but were trained to distinguish two different concentrations of the same odor (0.25% vs. 1.0% ethyl tiglate, *Figure 2c*). Silencing LEC again reduced the fraction of correct responses to the chance level (LED off: 77.40 ± 1.58%, LED on: 52.37 ± 2.10%, p = 0.0003, paired *t*-test, *n* = 6 mice), indicating that the LEC is also essential for detecting odor intensity and analysis of lick times revealed that mice could perform this discrimination rapidly (within 200 ms of odor inhalation, *Figure 2c*, *Figure 2—figure supplement 1*). Sham experiments confirmed that LED illumination alone could not produce these effects (*Figure 2—figure supplement 1*). Together, these findings indicate that LEC plays an essential role in behavior requiring discrimination of odor identity or intensity. Furthermore, the rapid speed of LEC-dependent behavior during single sniffs is entirely consistent with the rapid time course of odor-evoked activity we observe in L2.

## Ensemble coding of odor identity in LEC

We made acute recordings of L2 activity while mice were passively exposed to 11 odorants and analyzed responses of individual cells (*n* = 19 recordings in 17 mice) to examine odor identity coding (*Figure 3a*). Across all cells (*n* = 576), odor-evoked firing peaked in a time window 50–100 ms from the start of odor inhalation, considerably earlier than respiration-coupled activity observed in the absence of applied odor (blank response, *Figure 3b*). We thus used this 50-ms time window for analysis of rapid odor-evoked activity in LEC. Individual odors activated overlapping ensembles of L2 neurons

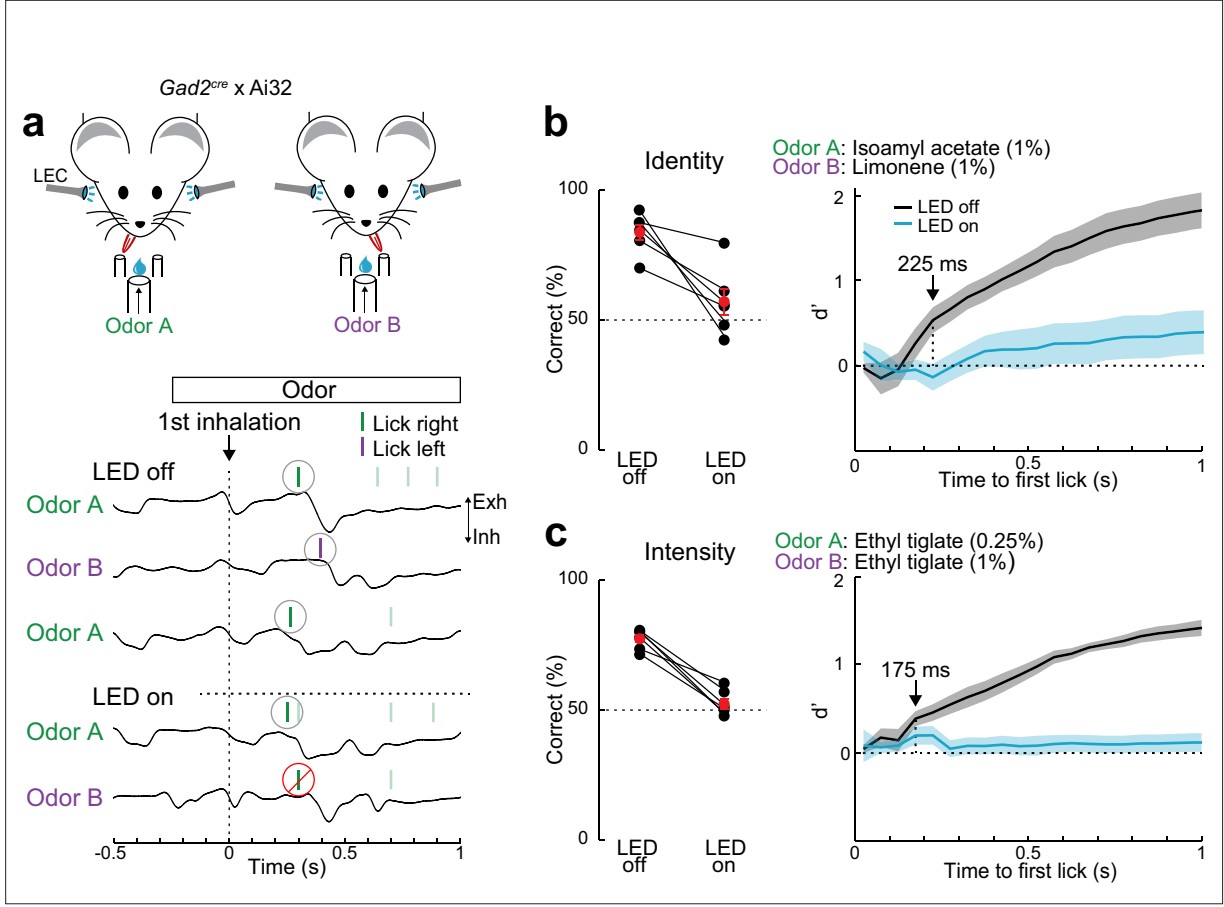

**Figure 2.** Lateral entorhinal cortex (LEC) is essential for rapid odor-driven behavior. (**a**) Top, experimental setup for two-alternative forced-choice (2AFC) odor-driven behavior with bilateral optogenetic silencing of the LEC in head-fixed mice expressing ChR2 in GABAergic neurons. Bottom, representative respiration traces aligned to the first inhalation of odor and superimposed lick responses (colored bars) during odor discrimination. (**b**) LEC is required for discrimination of odor identity. Left, percent of correct choices for mice trained to discriminate different odors (isoamyl acetate vs. limonene, both at 1.0%) on trials with (LED on) and without (LED off) optogenetic LEC silencing (*n* = 6 mice). Red circles, mean ± standard error of the mean (SEM). Right: *d'* of cumulative responses over time. Under control conditions (LED off), *d'* becomes significantly different 225 ms following odor inhalation. (**c**) LEC is required for discrimination of odor intensity. Left, percent of correct choices for mice trained to discriminate different concentrations of the same odor (0.25% vs. 1.0% ethyl tiglate) for trials with and without LEC silencing (*n* = 6 mice). Red circles, mean ± SEM. Right: *d'* of cumulative responses over time. Under control conditions (LED off), *d'* becomes significantly different 175 ms following odor inhalation. Error bars represent SEM.

The online version of this article includes the following source data and figure supplement(s) for figure 2:

**Source data 1.** Source data for *Figure 2*.

**Figure supplement 1.** Behavioral trial structure and performance in 2AFC discrimination tasks.

**Figure supplement 1—source data 1.** Source data for *Figure 2—figure supplement 1*.

with 48% of cells showing increases in firing rate to two or more odors (*Figure 3c*). Sorting responses of significantly activated cells normalized to the odors causing their strongest activation revealed that cells respond best to a subset of preferred odors (*Figure 3d*).

To study how activity in populations of L2 neurons might be used to encode odor identity, we used principal component analysis (PCA) of trial-by-trial response vectors from firing activity of all neurons recorded for each odor (see Methods). Vectors were binned in 1-ms intervals to examine the trajectory of the first three principal components following odor inhalation. Responses rapidly became more dispersed (discriminable) in PCA space (i.e., 0- vs. 75-ms postinhalation onset) and reverted to the preodor condition by 200 ms (*Figure 3e*). We measured discriminability using the Euclidean distance in PCA space for the different odors. This confirmed that the biggest difference in population activity occurred during the early peak of LEC odor-evoked activity (*Figure 3f*). We next trained a linear classifier to test the ability to decode odor identity from population activity. Using a sliding time window

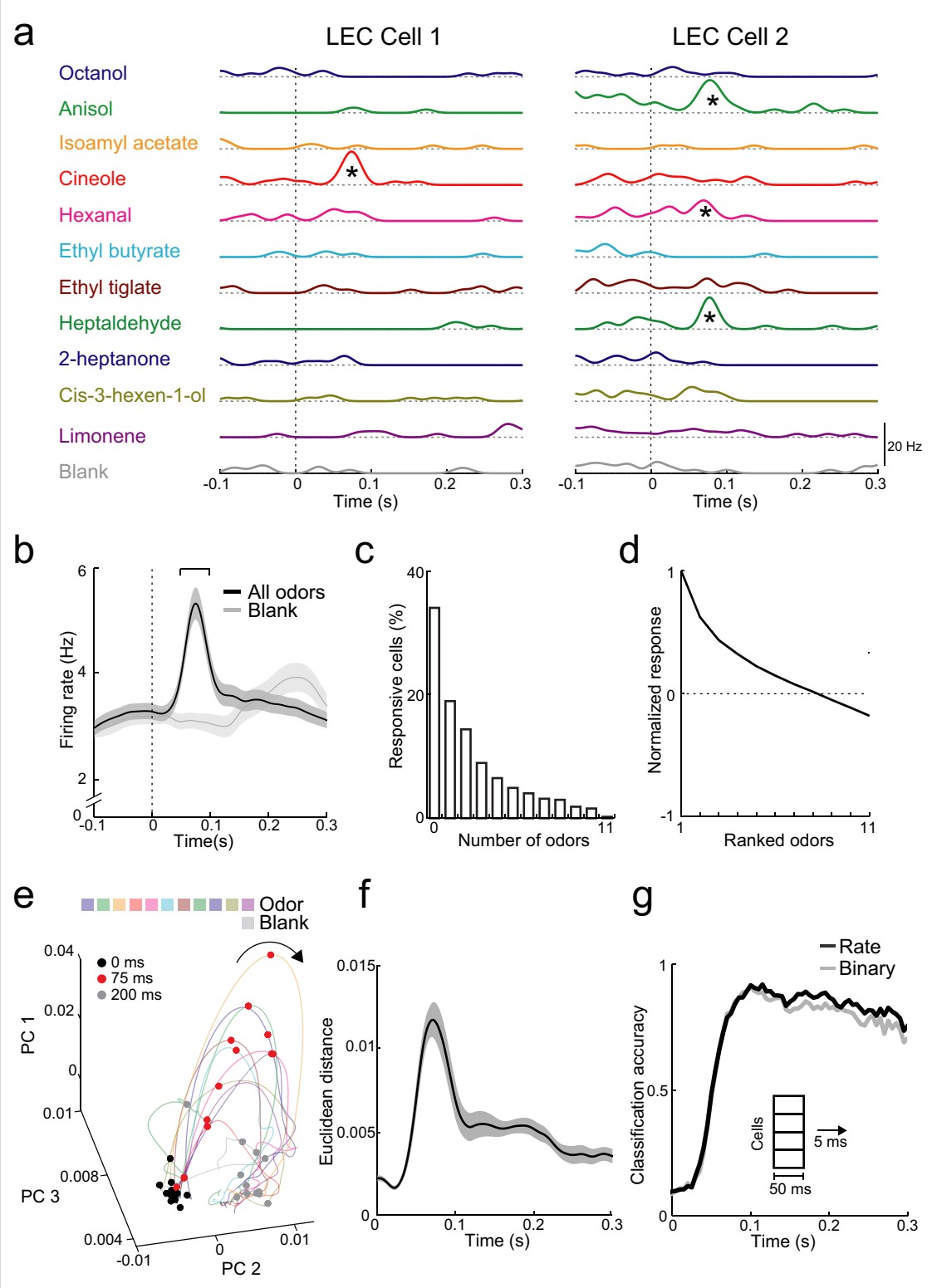

**Figure 3.** Rapid increases in firing rate encode odor identity in lateral entorhinal cortex (LEC) layer 2 (L2). (**a**) Spike-density functions of inhalation-aligned odor-evoked activity (11 odors and blank) for 2 representative LEC L2 cells. Asterisks, significant odor-evoked increases in firing rate. (**b**) Averaged spike-density function of activity in LEC L2 (*n* = 576 cells) shows early increase in firing rate for odor-evoked responses (black line) vs. baseline respiration-coupled activity (blank, gray line). Bracket, 50-ms window used to measure early odor-evoked firing. (**c**) Percent of cells responsive from 0 to 11 odors. (**d**) Normalized and ranked odor-evoked firing rate change relative to blank in LEC L2. (**e–g**) Principal component analysis reveals that LEC

*Figure 3 continued on next page*

*Figure 3 continued*

cell ensembles can rapidly distinguish odor identity. (**e**) Trajectories of the first three principal components of odor-evoked activity in LEC L2 over time following odor inhalation (each odor indicated by different colored line). Circles represent time points relative to odor inhalation: 0 ms (black), 75 ms (red), and 200 ms (gray). (**f**) Euclidean distance of the first three principal components (69% explained variance) between 11 odors (binned over 1-ms time window) shows discriminability peak within 100 ms of odor inhalation. (**g**) A linear classifier can rapidly discriminate odor identity from early odor-evoked changes in firing rate or binary categorization of activity. Classification accuracy is plotted using mean firing rate (black) or binary measure of cell activation (gray) using a 50-ms sliding window with 5-ms steps. Error bars represent standard error of the mean (SEM).

The online version of this article includes the following source data and figure supplement(s) for figure 3:

**Source data 1.** Source data for *Figure 3*.

**Figure supplement 1.** Classifier accuracy.

**Figure supplement 1—source data 1.** Source data for *Figure 3—figure supplement 1*.

(integrating 50 ms of activity every 5 ms), the classifier distinguished odor identities with high accuracy (>90 %) within 100 ms of odor inhalation (*Figure 3g*). We found an equally high accuracy in odor identification when cells were characterized only based on whether they showed significant increases in firing rate (*Figure 3g*). Thus, simply knowing which cell ensembles are active across the population provide as much information about odors as cell firing rates. While decoding accuracy remained high over the duration of the respiration cycle, reducing the integration time window for the linear classifier revealed a more transient increase in accuracy locked to the early phase of odor inhalation (*Figure 3—figure supplement 1*). This indicates that LEC population activity during later phases of individual sniffs best contains information about odor identity when activity is integrated over longer time windows. Classifier performance specifically during the rapid peak of odor-evoked activity (50- to 100-ms postinhalation) improved as the population of cells used for classification increased (*Figure 3—figure supplement 1*), however, high levels of accuracy (>50%) were readily achieved with ensembles of <250 L2 neurons.

## Spike timing in LEC encodes odor intensity

Does early odor-evoked activity in LEC also contain information regarding odor intensity? To address this question, we analyzed activity of single L2 neurons in response to four concentrations (0.25%, 0.33%, 0.50%, and 1.00%) of the same set of 11 odors. This fourfold change in concentration is within the range we used for behavioral discrimination (0.25% and 1.0%). Individual cells had diverse patterns of responses to different odor concentrations, with some showing a simple scaling of firing rate as odor concentration increased while others showed concentration-dependent changes in the time course of the evoked response (*Figure 4a*). Averaging odor-evoked activity across all units revealed only a weak increase in early firing rate for a fourfold change in odor concentration (50- to 100-ms postinhalation, 0.8-fold increase, $\rho = 0.3$, p = 0.03, Spearman's rank correlation, *Figure 4b*, *Figure 4—figure supplement 1*). Similarly, we found only a modest increase in the percentage of activated cells per odor as concentration increased ($\rho = 0.3$, p = 0.05, Spearman's rank correlation, *Figure 4—figure supplement 1*) and lifetime sparseness (a measure of response selectivity) was weakly sensitive to changes in concentration ($\rho = 0.04$, p = 0.04, Spearman's rank correlation, *Figure 4—figure supplement 1*). These findings suggest that odor intensity might not be optimally represented by the firing rates of ensembles of LEC neurons.

We used PCA of the pseudopopulation activity grouped by odor concentration to understand how responses evolved over the time course of single sniffs. Interestingly, responses to the four concentrations rapidly dispersed in PCA space after odor inhalation and returned to preodor levels of variability within 200 ms (*Figure 4c*). This suggests that ensemble activity based on early odor-evoked changes in firing rate could be used to discriminate odor intensity. Indeed, the Euclidean distances in PCA space between responses to different odor concentrations revealed a clear peak within 100 ms of odor inhalation (*Figure 4d*). However, these values were much less than those obtained from PCA comparing different odors and increasing concentration only slightly increased the distances that distinguished between odors (*Figure 4d*). Thus, population activity showed smaller differences (less discriminability) between odor concentrations when compared to activity representing different odor identities.

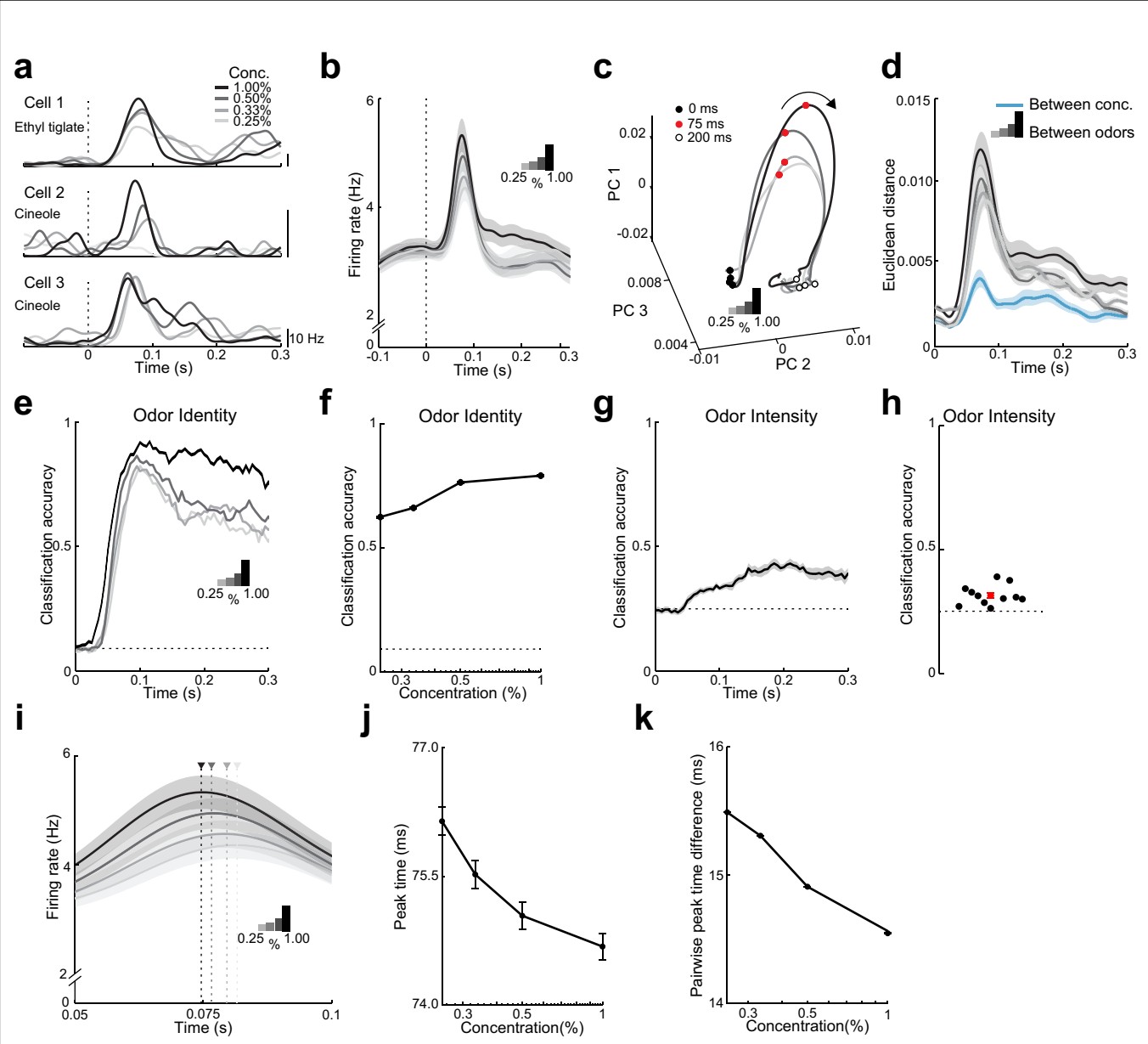

**Figure 4.** Both rate and timing of odor-evoked activity encode odor intensity in lateral entorhinal cortex (LEC) layer 2 (L2). (**a**) Spike-density functions of odor-evoked activity for three representative LEC L2 cell–odor pairs at four concentrations indicated by line shading (0.25%, 0.33%, 0.50%, and 1.00%). (**b**) Averaged spike-density function of odor-evoked activity aligned to the first inhalation across four concentrations of the same odors for the LEC L2 cell population (*n* = 576 cells, 11 odors at 0.25%, 0.33%, 0.50%, and 1.00%). (**c, d**) Principal component analysis of the cell population for responses to four odor concentrations indicates that firing rate can be used to discriminate odor intensity, but discriminability is poorer than that for odor identity. (**c**) Trajectories of the first three principal components of odor-evoked activity over time following odor inhalation (each odor concentration indicated by a different line shading). Circles represent time points relative to odor inhalation: 0 ms (filled black), 75 ms (red), and 200 ms (open black). (**d**) Euclidean distance of the first three principal components of odor-evoked activity in the LEC between responses of the four concentrations (blue) shows a peak in discriminability within 100 ms of odor inhalation. The discriminability between responses of the same cells to 11 odors for the four different intensities (gray to black) is much higher. Responses are binned with a 1-ms time window. (**e–h**) A linear classifier shows that odor identity discrimination somewhat improves as odor concentration increases and that odor intensity can be weakly discerned from firing rate changes. (**e**) Classification accuracy for odor identity at the four concentrations indicated by the shaded lines using a 50-ms sliding window with 5-ms steps. (**f**) Classification accuracy of odor identity at different concentrations based on average firing rate in a fixed window 50–100 ms following odor inhalation. (**g**) Classification accuracy of odor intensity for each odor based on average firing rate using a sliding window. (**h**) Classification accuracy of odor intensity for each odor based on average firing rate in a fixed window 50–100 ms following odor inhalation. (**i–k**) Spike timing effectively encodes odor intensity. (**i**) Blow-up of data from (**b**) shows shift in timing of peak responses with odor concentration. Arrowheads mark peak times at the four concentrations. (**j**) Peak time of odor-evoked

*Figure 4 continued on next page*

*Figure 4 continued*

activity averaged across all cell–odor pairs shifts with changes in odor concentration. (**k**) Mean pairwise peak time difference of odor-evoked activity for cell–odor pairs indicates that spike firing becomes more synchronized as odor concentration increases. Error bars represent standard error of the mean (SEM).

The online version of this article includes the following source data and figure supplement(s) for figure 4:

**Source data 1.** Source data for *Figure 4*.

**Figure supplement 1.** Summary data and classifier accuracy describing effects odor concentration on responses in lateral entorhinal cortex (LEC) layer 2 (L2).

**Figure supplement 1—source data 1.** Source data for *Figure 4—figure supplement 1*.

We trained a linear classifier to detect odor identity for the different concentrations using firing rates integrated over a sliding 50-ms window. Classification accuracy of odor identity only slightly improved as concentration increased (*Figure 4e, f*, *Figure 4—figure supplement 1*). In contrast, when we trained the linear classifier to detect odor concentration based on firing rate, classification accuracy was considerably worse (*Figure 4g*, *Figure 4—figure supplement 1*). Indeed, classification accuracy for each of the 11 tested odors was only marginally above chance level (*Figure 4h*). These results suggest that while L2 population firing rates can be used to discern both odor identity and intensity, rate coding appears relatively weaker at representing information regarding odor intensity.

What other properties of evoked activity might underlie the representation of odor intensity in LEC? When studying odor-evoked responses at higher temporal resolution, we found that the average time to peak firing of L2 cells shifted earlier as odor concentration increased (*Figure 4i*). We quantified this shift by determining the time of peak firing (relative to odor inhalation) for all cell–odor pairs across odor concentrations and found that peak times shifted earlier with increasing odor concentrations ($\rho$ = −0.04, p < 0.001, Spearman's rank correlation, *Figure 4j*). We next considered the possibility that changes in spike timing across the L2 cell population altered the relative synchrony of odor-evoked activity in LEC. Indeed, pairwise peak time differences for odor responses became smaller as odor concentration increased (*Figure 4k*, *Figure 4—figure supplement 1*). This indicates that activity in LEC becomes more synchronized with increases in odor concentration. Together, these findings show that odor intensity can be distinguished using a temporal code in LEC.

## Differential connectivity of fan and pyramidal cells with long range and local circuits

How do the two types of L2 principal neurons contribute to odor processing in LEC? We first used brain slices to study the functional properties of L2 fan and pyramidal cells. We took advantage of the fact that the two cell types can be distinguished by their distinct axonal projection patterns: fan cells project to DG granule cells while L2 pyramidal cells send feedback projections to the OB (*Chapuis et al., 2013*; *Leitner et al., 2016*; *Vandrey et al., 2020*). We injected green retrobeads into either the DG or OB and made targeted whole-cell current clamp recordings from bead-labeled cells in coronal LEC slices. In a subset of experiments, we used mice that express red fluorescent protein in olfactory cortical pyramidal cells (*Ntsr1(209)$^{cre}$* × Ai14, *Boyd et al., 2012*). Biocytin was added to the pipette internal solution for post hoc reconstruction of dendritic arbors. DG bead injection labeled neurons with cell bodies localized in L2a (*Figure 5a*) that possessed extensive, fan-like apical dendrites but few basal dendrites (*Figure 5b*, *Figure 5—figure supplement 1*). In contrast, OB injection labeled cells in L2b (*Figure 5c*) with typical pyramidal cell dendritic morphology (*Figure 5d*, *Figure 5—figure supplement 1*). Similar to previous studies of L2 fan and pyramidal cells (*Leitner et al., 2016*; *Tahvildari and Alonso, 2005*; *Vandrey et al., 2020*), current injection revealed differences in membrane excitability including resting potential (fan: −64 ± 1.1 mV [*n* = 15], pyramidal: −69.4 ± 1.7 mV [*n* = 10], p = 0.016, *t*-test), input resistance (fan: 147.6 ± 10.4 mOhm, pyramidal: 102.1 ± 12.8 mOhm, p = 0.012), and sag (fan: 3.7 ± 0.4 mV, pyramidal: 1.1 ± 0.4 mV, p = 0.0003).

We next used an optogenetic approach to examine whether fan and pyramidal cells differed in their olfactory input. To probe direct input from the OB, we took advantage of *Tbx21$^{cre}$* mice crossed with the Ai32 reporter line to selectively express ChR2 in OB mitral cells (*Haddad et al., 2013*). We used LEC slices from these mice to make simultaneous voltage-clamp recordings (−70 mV) from L2a fan and L2b pyramidal cells (*Figure 5e*) that were confirmed by post hoc anatomical reconstruction.

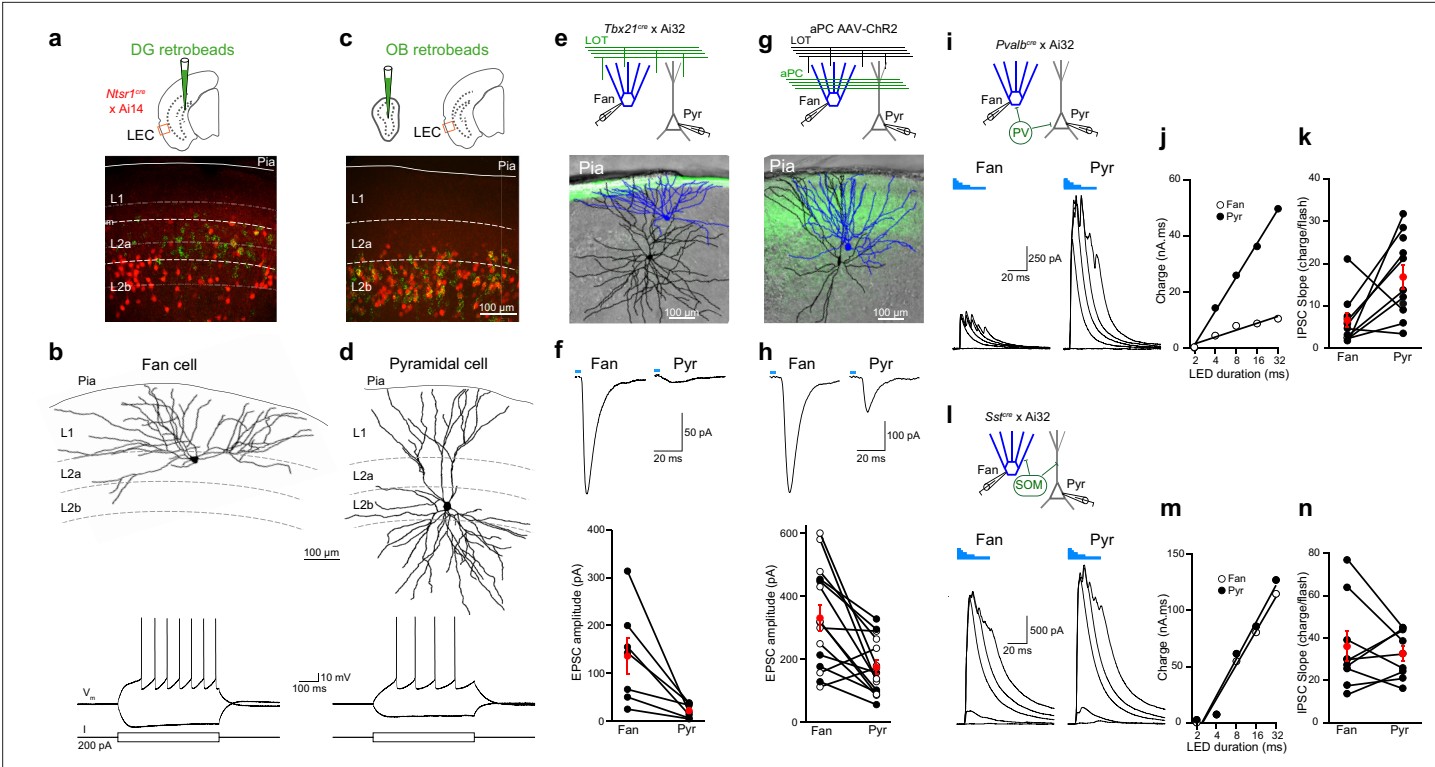

**Figure 5.** Layer 2 (L2) fan and pyramidal cells differ in olfactory input and inhibitory circuit connectivity. (**a**) Top, schematic of retrobead injection into the DG of *Ntsr1(209)^cre* mice expressing TdTomato. Bottom, image of lateral entorhinal cortex (LEC) region outlined in schematic showing DG-projecting cells (green) located in L2a and Ntsr1-positive cells (red) concentrated in L2b. (**b**) Targeted brain slice recording from a DG-projecting fan cell. Top, reconstruction of fan cell dendritic arbor. Bottom, membrane potential response to current steps. (**c**) Top: schematic of retrobead injection into the olfactory bulb (OB) of *Ntsr1(209)^cre* mice expressing TdTomato. Bottom, image of LEC region showing OB-projecting cells (green) and Ntsr1-positive cells (red) colocalize in L2b. (**d**) Targeted brain slice recording from an OB-projecting pyramidal cell. Top, reconstruction of pyramidal cell dendritic arbor. Bottom, membrane potential response to current steps. (**e, f**) Fan cells receive stronger direct OB input than pyramidal cells. (**e**) Top, recording schematic for paired recordings of fan and pyramidal cells in LEC slices expressing ChR2 in lateral olfactory tract (LOT) fibers. Bottom, reconstruction of simultaneously recorded fan cell (blue) and pyramidal cell (black) overlaid on image of LEC slice showing ChR2 expression in LOT (green). (**f**) Top, LED-evoked excitatory postsynaptic currents (EPSCs) in fan and pyramidal cells from e. Blue bar, LED flash. Bottom, summary of EPSC amplitudes elicited by stimulation of OB input. Black circles, individual cells. Red circles, mean ± standard error of the mean (SEM). (**g, h**) Fan cells receive stronger direct PCx input than pyramidal cells. (**g**) Top, recording schematic for paired recordings in LEC slices expressing ChR2 in PCx. Bottom, reconstruction of simultaneously recorded fan cell (blue) and pyramidal cell (black) overlaid on image of LEC slice showing ChR2 expression in PCx fibers (green). (**h**) Top, LED-evoked EPSCs in fan and pyramidal cells from (g). Blue bar, LED. Bottom, summary of EPSC amplitudes elicited by stimulation of PCx input. Black circles, cells in control artificial cerebrospinal fluid (aCSF). Open circles, cells recorded in tetrodotoxin (1 µM) and 4-aminopyridine (100 µM). Red circles, mean ± SEM. (**i–k**) Fan cells receive less parvalbumin (PV) cell-mediated inhibition than pyramidal cells. (**i**) Top, recording schematic. Bottom, LED-evoked inhibitory postsynaptic currents (IPSCs) elicited in simultaneously recorded fan and pyramidal cells. Traces indicate responses to five LED pulses of increasing duration (blue bars). (**j**) Inhibitory charge for fan cell (open circles) and pyramidal cell (filled circles) shown in (i). Lines fit to points are used to measure slope of input–output relationship. (**k**) Summary of IPSC slopes for fan and pyramidal cells in response to activation of PV cells. Black circles, individual cells. Red circles, mean ± SEM. (**l–n**) Fan cells and pyramidal cells receive similar somatostatin (SOM) cell-mediated inhibition. (**l**) Top, recording schematic. Bottom, LED-evoked IPSCs elicited in simultaneously recorded fan and pyramidal cells. Traces indicate responses to five LED pulses of increasing duration (blue bars). (**m**) Inhibitory charge for fan cell (open circles) and pyramidal cell (filled circles) shown in (l). (**n**) Summary of IPSC slopes for fan and pyramidal cells in response to activation of SOM cells. Black circles, individual cells. Red circles, mean ± SEM.

The online version of this article includes the following source data and figure supplement(s) for figure 5:

**Source data 1.** Source data for *Figure 5*.

**Figure supplement 1.** Dendritic arbors, paired-pulse ratio, and minimal stimulation for fan and pyramidal cells in lateral entorhinal cortex (LEC).

**Figure supplement 1—source data 1.** Source data for *Figure 5—figure supplement 1*.

Excitatory postsynaptic currents (EPSCs) evoked from ChR2-expressing LOT fibers (473 nm, 4-ms flash) were significantly larger in fan vs. pyramidal cells (*Figure 5f*, 136.7 ± 3 and 22.3 ± 5.5 pA, respectively, *n* = 7 pairs, p = 0.02, paired *t*-test). Paired-pulse facilitation (200-ms interstimulus interval) was identical for fan and pyramidal cell EPSCs (*Figure 5—figure supplement 1*, paired pulse ratio 1.9 ± 0.2 and 1.8 ± 0.3, respectively, p = 0.69, paired *t*-test) suggesting that LOT release probability was the same onto both cell types. Moreover, the amplitude of single fiber EPSCs recorded in fan (*n* = 8) and pyramidal (*n* = 7) cells using minimal LOT stimulation were also virtually identical (*Figure 5—figure supplement 1*, 9.7 ± 1.7 and 7.9 ± 1.1 pA, respectively, p = 0.35, *t*-test). These results indicate that the stronger LOT-evoked responses of fan cells reflect the fact that they receive inputs from more OB mitral cells than pyramidal cells. Projections from piriform cortex provide another source of olfactory input to the LEC. To study these inputs, we used LEC slices from mice injected in anterior PCx with adeno-associated virus (AAV) driving expression of ChR2. A subset of experiments were performed in the presence of tetrodotoxin (1 µM) and 4-aminopyridine (100 µM) to ensure monosynaptic input (*Petreanu et al., 2009*). Simultaneous recordings revealed larger ChR2-evoked EPSCs in fan vs. pyramidal cells (*Figure 5g, h*, 331 ± 41.4 and 174 ± 22.6 pA, respectively, *n* = 15 pairs, p = 0.002, paired *t*-test) indicating that fan cells also receive stronger olfactory input relayed from the piriform cortex.

In addition to olfactory excitatory input, we considered the possibility that fan and pyramidal cells might differ in their inhibition by local GABAergic interneurons. We addressed this by studying the impact of two major classes of inhibitory interneurons: soma-targeting, parvalbumin (PV) cells and dendrite-targeting somatostatin (SOM) cells (*Isaacson and Scanziani, 2011*). We made simultaneous recordings of inhibitory postsynaptic currents (IPSCs, $V_m$ = 0 mV) from fan and pyramidal cells using LEC slices from *Pvalb^cre* (*Figure 5i–k*) or *Sst^cre* mice (*Figure 5l–n*) crossed with a ChR2 reporter line (Ai32). We drove interneuron firing with different durations of LED illumination (2–32 ms) and fit the relationship between LED duration and inhibitory charge with a line to compare the slopes of the input/output relationship for each cell pair. Interestingly, these experiments revealed that fan cells received substantially less inhibition from PV cells than pyramidal cells (*Figure 5i–k*) while both cell types received the same relative amount of SOM cell-mediated inhibition (*Figure 5l–n*). Thus, fan and pyramidal cells also differ in their connectivity with local interneuron circuits.

## Temporal shift of pyramidal cell firing relative to fan cells regulates synchronous activity

The differences in olfactory input and PV cell-mediated inhibition between L2 fan and pyramidal cells suggest that the two principal cells may respond heterogeneously to odors. To investigate this idea, we identified cre mouse lines that could be used to target in vivo recordings to the different L2 principal cell types. We found that *NetrinG1^cre* mice, which selectively label semilunar principal cells in PCx (*Bolding et al., 2020*), specifically targeted fan cells in LEC. In *NetrinG1^cre* × Ai14 mice, TdTomato-expressing cells localized to L2a and were colabeled by retrobeads injected in the DG (*Figure 6a*, 90.47 ± 1.47%, *n* = 6 slices from three mice). Furthermore, immunolabeling revealed that *NetrinG1^cre* expression was restricted to reelin-positive neurons in LEC L2a and did not overlap with L2b cells expressing calbindin (*Figure 6—figure supplement 1*). To target recordings to L2 pyramidal cells, we used *Calb1^cre* mice (*Leitner et al., 2016*). We confirmed that cells expressing *Calb1^cre* were concentrated in L2b, colabeled by retrobeads injected in the OB (81.89 ± 2.39%, *n* = 6 slices from three mice), and immuno-positive for calbindin but not reelin (*Figure 6b*, *Figure 6—figure supplement 1*). Consistent with previous histological work describing LEC calbindin-expressing pyramidal cells projecting to CA1 (*Ohara et al., 2019*), we found small numbers of *Calb1^cre* cells in layer 3. Although calbindin is also expressed in some GABAergic neuron subtypes, immunolabeling in *Gad2^cre* × Ai14 mice revealed very few calbindin-positive interneurons in LEC (*Figure 6—figure supplement 1*).

We crossed *NetrinG1^cre* and *Calb1^cre* mice with the Ai32 line to express ChR2 in LEC L2 fan and pyramidal cells, respectively, and used phototagging (*Lima et al., 2009*) to identify the cell types during single-unit recording in awake, head-fixed mice (*n* = 12 recordings from 10 *NetrinG1^cre* mice and *n* = 7 recordings from 7 *Calb1^cre* mice, *Figure 6—figure supplement 1*). Both identified cell types showed an early peak of odor-evoked activity (*Figure 6c, d*) and we found only modest differences in ensemble coding of odor identity; the number of activated units per odor and mean evoked firing rate was slightly greater for pyramidal cells while lifetime sparseness (a measure of odor selectivity) was higher for fan cells (*Figure 6—figure supplement 1*).

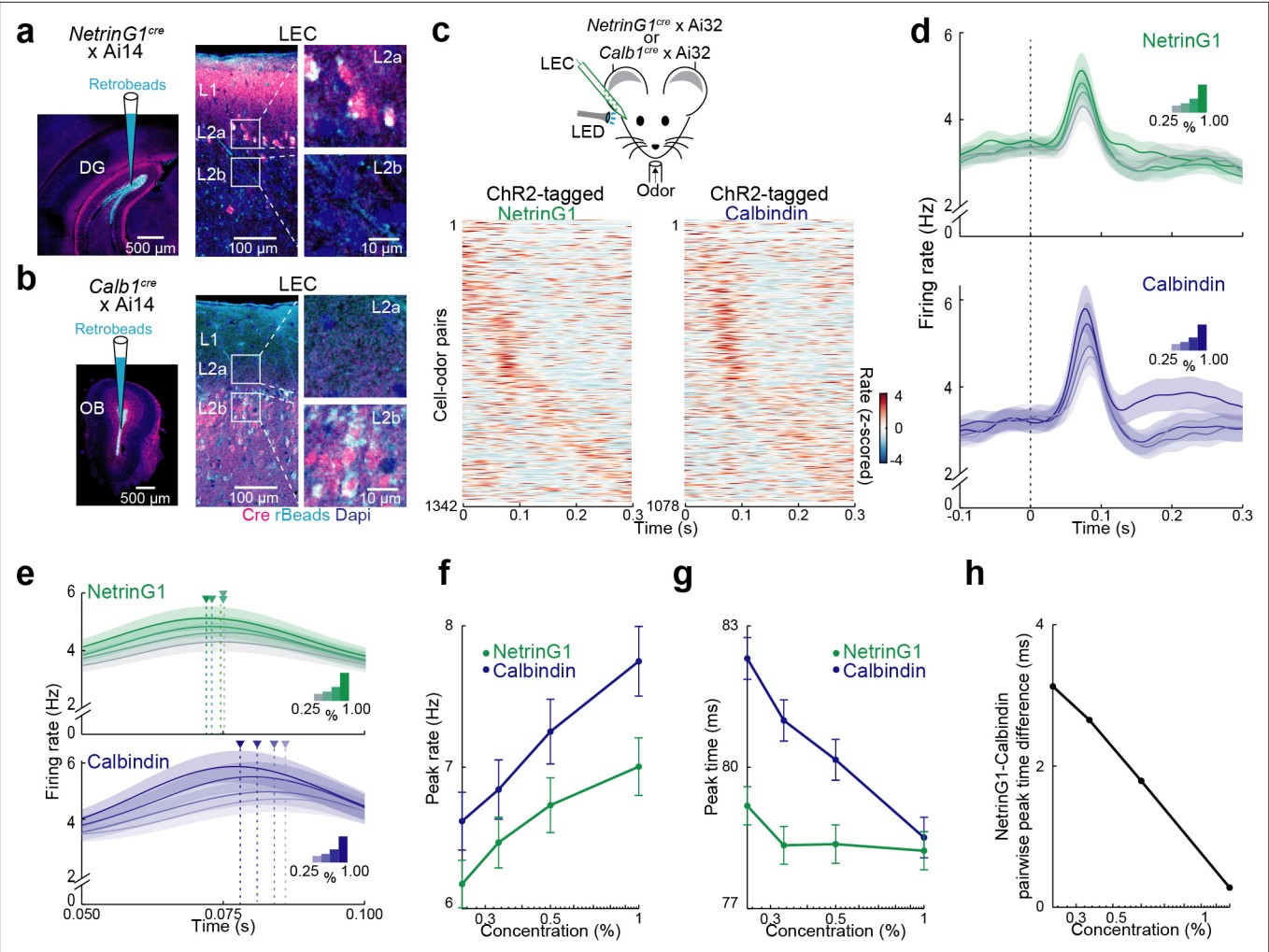

**Figure 6.** Temporal coding of odor intensity by fan and pyramidal cells. (**a, b**) Transgenic mouse lines allow targeting of layer 2 (L2) fan and pyramidal cells in lateral entorhinal cortex (LEC). (**a**) *NetrinG1*-cre expression in L2a fan cells. Left, site of DG retrobead (blue) injection in *NetrinG1cre* × Ai14 mouse. Right, LEC section indicates cre restricted to DG-projecting cells in L2a. (**b**) *Calb1*-cre expression in L2b pyramidal cells. Left, site of olfactory bulb (OB) retrobead (blue) injection in *Calb1cre* × Ai14 mouse. Right, LEC section indicates cre within OB-projecting cells in L2b. (**c**) Odors evoke rapid responses in optogenetically tagged fan and pyramidal cells. Top, experimental setup. Bottom, plots of odor-evoked firing rates for all cell–odor pairs rank ordered by time of peak firing for ChR2-tagged NetrinG1 (left) and calbindin (right) neurons (*n* = 122 and 98 cells, respectively). Peristimulus time histograms using only even trials are sorted relative to the peaks obtained from odd trials. (**d**) Odor concentration-dependent timing of pyramidal cell activity. Average time course of activity evoked by different odor concentrations (0.25%, 0.33%, 0.50%, and 1.00%) for tagged NetrinG1/fan (top) and calbindin/pyramidal (bottom) cells. (**e**) Traces in (**d**) shown on a faster timescale. Arrowheads mark times of peak firing for the different intensities. (**f**) Peak rates of odor-evoked activity at different intensities for NetrinG1/fan and calbindin/pyramidal cell–odor pairs. (**g**) Peak times of odor-evoked activity at different intensities reveal strong dependence on odor concentration for calbindin/pyramidal but not NetrinG1/fan cells. (**h**) Synchrony between fan and pyramidal cell firing depends on odor concentration. Pairwise peak time differences of NetrinG1/fan and calbindin/pyramidal cell–odor pairs decrease as odor concentration increases. Error bars represent standard error of the mean (SEM).

The online version of this article includes the following source data and figure supplement(s) for figure 6:

**Source data 1.** Source data for *Figure 6*.

**Figure supplement 1.** Expression specificity for NetrinG1-cre and Calbindin-cre mice and optotagging of princpal neurons in lateral entorhinal cortex (LEC).

**Figure supplement 1—source data 1.** Source data for *Figure 6—figure supplement 1*.

In contrast to the subtle distinctions in odor identity coding, we found a notable difference in how changes in odor concentration modulated spike timing in fan and pyramidal cells. While the average latency to peak response shifted earlier as concentration increased for pyramidal cells, the time course of fan cell responses appeared concentration invariant (*Figure 6e*). We confirmed this by quantifying

the peak responses for all cell–odor pairs. We found that although peak firing rates slightly increased with odor concentration for both cell types (*Figure 6f*, fan: $\rho$ = 0.04, p = 0.009, $n$ = 112 cells; pyramidal: $\rho$ = 0.04, p = 0.009, $n$ = 98 cells, Spearman's rank correlation), they differed significantly in how odor concentration modulated spike timing: time to peak firing for pyramidal cells became faster as concentration increased, whereas fan cell peak firing times were largely unaffected (*Figure 6g*, fan: $\rho$ = −0.02, p = 0.2; pyramidal: $\rho$ = −0.09, p < 0.001, Spearman's rank correlation; fan vs. pyramidal peak time: 0.25% p < 0.001, 0.33% p < 0.001, 0.5% p = 0.01, 1% p = 0.7, *t*-test). Interestingly, the peak firing times of fan cells typically preceded pyramidal cells (*Figure 6g*) and pyramidal, but not fan cells, fired more synchronously as odor concentration increased (*Figure 6—figure supplement 1*). The temporal shift in pyramidal cell activity and earlier (but constant) timing of fan cell responses underlies spike synchronization between the cell types as odor concentration increases (*Figure 6h*, $\rho$ = −0.05, p < 0.001, Spearman's rank correlation). Together, these results show that relative differences in spike timing between fan and pyramidal cells can account for the temporal coding of odor intensity in LEC L2.

## A purely temporal code for odor intensity in CA1

Spike timing and synchrony in L2 of LEC provide a potential mechanism for odor intensity coding, but is this information relevant downstream in the hippocampus? To probe how information from LEC was routed to the hippocampal formation, we first traced the projections from L2 principal cells by injecting AAV1-CAG-FLEX-tdTomato into the LEC of *NetrinG1*$^{cre}$ and *Calb1*$^{cre}$ mice (*Figure 7a, b*). This showed that fan cells send dense input to the outer molecular layer of the DG via the lateral perforant path (*Figure 7a*, *Leitner et al., 2016*; *Vandrey et al., 2020*). Injections in *Calb1*$^{cre}$ mice revealed projections from L2 pyramidal cells to hippocampal CA1 through the temporoammonic pathway (*Figure 7b*, *Amaral and Witter, 1989*; *Kitamura et al., 2014*; *Masurkar et al., 2017*). This tracing confirmed that information from LEC L2 is relayed to CA1 via both direct (pyramidal cell) and indirect (fan cell) pathways.

We next used linear silicon probes to record odor-evoked activity in intermediate CA1 in awake, head-fixed mice ($n$ = 609 cells from 13 recordings in 11 mice). Like PCx and LEC, cells in CA1 preferentially responded early during individual sniffs (*Figure 7c, d*) and peak odor-evoked firing in CA1 (106 ms, odor concentration 1%) occurred briefly after LEC fan (72 ms) and pyramidal (78 ms) cells. Surprisingly, in contrast to LEC L2, the peak firing rate in CA1 was insensitive to odor concentration (*Figure 7d* compared to *Figure 4b*). Unlike LEC, the percentage of activated cells per odor, mean firing rate, and response sparseness were also concentration independent in CA1 (*Figure 7—figure supplement 1*). This suggests that rate coding features found in LEC that could contribute to odor intensity discrimination are discarded at the level of CA1. Indeed, although PCA (*Figure 7—figure supplement 1*) and the use of linear classifiers revealed that odor identity could be distinguished with high accuracy (*Figure 7e*, *Figure 7—figure supplement 1*), odor intensity could not be decoded from CA1 firing rates (*Figure 7f*, *Figure 7—figure supplement 1*). Thus, firing rates contain information about odor identity, but not intensity in CA1.

Importantly, we observed that the timing of early odor-evoked activity was dependent on concentration in CA1 (*Figure 7g*). Although peak firing rate was concentration insensitive, the time to peak shifted earlier as concentration increased (*Figure 7h, i*, peak rate: $\rho$ = −0.002, p = 0.7; peak time: $\rho$ = −0.04, p < 0.001, $n$ = 609 cells, Spearman's rank correlation). As we found for LEC, pairwise peak time differences were reduced with increasing odor concentration ($\rho$ = −0.01, p < 0.001, Spearman's rank correlation), indicating that firing also became more synchronized within CA1 (*Figure 7j*, *Figure 7—figure supplement 1*). These findings indicate that downstream of LEC, odor intensity must ultimately be represented by changes in spike timing in CA1.

## Discussion

In this study, we examined olfactory information processing in the LEC of awake mice. We found that odors activate principal L2 cells early during individual sniffs and that LEC is essential for rapid odor-guided behavior. Odor identity was readily decoded from the firing rates of ensembles of L2 cells, but rate coding appeared less useful for discerning odor intensity. Interestingly, changes in odor concentration were well described by shifts in the timing and synchrony of L2 cell spikes. We found that the

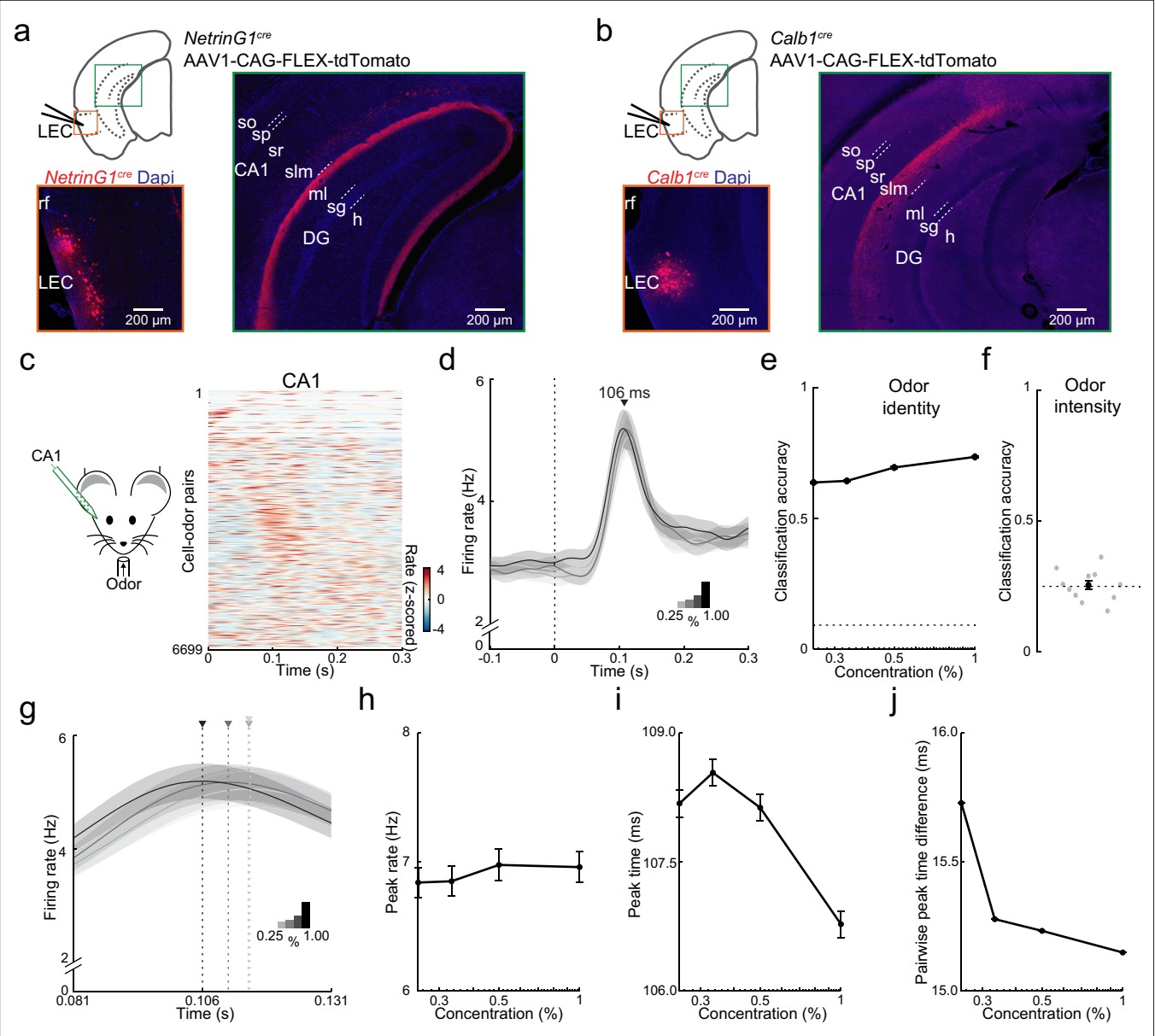

**Figure 7.** Purely temporal coding of odor intensity in CA1. (**a, b**) Fan and pyramidal cells target distinct hippocampal regions. (**a**) Schematic of fan cell anterograde viral tracing strategy in *NetrinG1cre* mice and images showing AAV-FLEX-tdTomato expression (red) at lateral entorhinal cortex (LEC) injection site (left) and projections concentrated in DG (right). (**b**) Schematic of pyramidal cell tracing strategy in *Calb1cre* mice and images showing AAV-FLEX-tdTomato expression (red) at LEC injection site (left) and projections concentrated in CA1 stratum (s) lacunosum molecular (slm, right). Hippocampal regions: so, s. oriens; sp, s. pyramidale; sr, s. radiatum; ml, DG molecular layer; sg, DG s. granulosum; h, DG hilus. (**c–f**) Rapid odor-evoked changes in firing rate encode odor identity but not intensity in CA1 pyramidal cells. (**c**) CA1 pyramidal cells respond rapidly to odors. Left, experimental setup. Right, odor-evoked firing rates for all cell–odor pairs rank ordered by time of peak firing in CA1 (*n* = 609 cells, 11 odors). Peristimulus time histograms using only even trials are sorted relative to the peaks obtained from odd trials. (**d**) Average activity at four odor concentrations for CA1 cells aligned to inhalation onset show that firing rate is independent of odor concentration. (**e**) Firing rate encodes odor identity in CA1. Plot shows classification accuracy of odor identity at the different intensities (0.25%, 0.33%, 0.50%, and 1.00%) based on the average firing rate in a fixed time window (81–131 ms after inhalation onset). Chance, dotted line. (**f**) Firing rate does not encode odor intensity. Classification accuracy of odor intensity for 11 odors based on average firing rate in the same fixed time is no better than chance (dotted line). (**g–j**) Odor intensity encoded by spike timing changes in CA1. (**g**) Data from (**d**) on a faster timescale show odor concentration-dependent shift in timing of peak response. Arrowheads indicate peak times for the concentrations indicated by different shading. (**h**) Peak firing rate of odor-evoked activity for all cell–odor pairs is independent of concentration. (**i**) Peak time of odor-evoked activity for cell–odor pairs is concentration dependent. (**j**) Pairwise peak time differences between neurons indicates that odor-evoked firing in CA1 becomes more synchronous as odor concentration is increased. Error bars represent standard error of the mean (SEM).

*Figure 7 continued on next page*

*Figure 7 continued*

The online version of this article includes the following source data and figure supplement(s) for figure 7:

**Source data 1.** Source data for *Figure 7*.

**Figure supplement 1.** Summary data and classifier accuracy describing effects odor concentration on responses in hippocampal CA1.

**Figure supplement 1—source data 1.** Source data for *Figure 7—figure supplement 1*.

---

two types of L2 principal neurons, fan and pyramidal cells, differ in their functional properties. Slice recordings revealed stronger olfactory input and weaker PV cell inhibition of L2 fan cells compared to pyramidal cells and targeted in vivo recordings showed that spike timing differences between the cell types contribute to odor intensity coding in LEC. Interestingly, we found that CA1 discards any concentration-dependent firing rate changes inherited from LEC and represents odor intensity using a purely temporal code.

In awake animals, the timing of respiration-coupled activity driven by odors varies across brain regions. While population activity of OB mitral cells spans individual respiration cycles, L2/3 pyramidal cells in PCx respond to odors preferentially early after odor inhalation (*Bolding and Franks, 2018*; *Shusterman et al., 2011*). Recurrent inhibition has been found to enforce the short time window for sniff-coupled activity elicited by odors in PCx (*Bolding and Franks, 2018*). Here, we show in LEC, which receives input from both the OB and PCx, that odor-evoked activity is also largely limited to an early window within 100 ms of odor inhalation. Like PCx, local feedback inhibition presumably curtails late odor-evoked activity in LEC.

Previous behavioral studies have differed regarding the importance of LEC for odor discrimination. Chemical inactivation using LEC muscimol infusion in rats was reported to impair discrimination of complex but not simple multiodor mixtures (*Chapuis et al., 2013*). This suggests that LEC is only required for behavior requiring difficult odor discrimination. In contrast, more recent work in mice found that LEC muscimol infusion (*Lee et al., 2021*) or optogenetic suppression of LEC inputs to CA1 (*Li et al., 2017*) degraded performance in simple, two-odorant discrimination tasks. Moreover, the LEC and fan cells in particular appear to play a critical role in the learning of new olfactory cue–reward associations (*Igarashi et al., 2014*; *Lee et al., 2021*). We found that optogenetic suppression of the LEC abolished rapid behavioral discrimination of both odor identity and intensity, indicating that the LEC is essential for simple odor-guided behavior. Mice made decisions during the 2AFC task using information obtained early during single sniffs of odor. Thus, it is likely that L2 activity early during individual sniffs is critical for odor-driven behavior. Our experiments required simple discrimination of markedly different odorants or odor concentrations; we do not rule out the possibility that mice could integrate LEC activity over longer periods (even across multiple sniffs) during tasks requiring more difficult odor discrimination.

While acute optogenetic silencing of LEC disrupted performance in behavioral tasks requiring discrimination of odor identity and intensity, our findings do not necessarily prove that odor discrimination within the LEC is required for behavior. It could be the case that LEC is essential for the retrieval of learned odor–reward associations while discrimination is provided elsewhere (i.e., piriform cortex). Nonetheless, our results are consistent with LEC playing a critical role in odor-driven behavior. An additional caveat to experiments using optogenetic silencing, is that acute perturbations could disrupt behavior by removing 'permissive' activity for downstream structures, rather than implicating the target circuit as 'instructive' for the behavior (*Wolff and Ölveczky, 2018*). Future experiments examining the effects of chronic LEC inactivation on odor discrimination will be useful to probe this possibility.

Given the timing of odor-guided behavior and odor-evoked activity in LEC, we studied how information regarding odor identity is encoded by rapid L2 cell responses. We find that different odors are represented by overlapping ensembles of active cells in LEC, similar to odor coding in PCx (*Blazing and Franks, 2020*; *Miura et al., 2012*; *Poo and Isaacson, 2009*; *Stettler and Axel, 2009*; *Uchida et al., 2014*). Population analyses established that a firing rate code distinguished odor identity in L2 and distinct odor representations emerged rapidly (<100 ms) after inhalation. Interestingly, as in PCx (*Bolding and Franks, 2017*), simply knowing whether cells were activated or not by odors was as effective as knowing actual firing rates when decoding odor identity from population activity.

**eLife** Research article

Increases in odor concentration over a fourfold range caused modest increases in firing rate of L2 cells and population analyses revealed that increases in concentration could improve discrimination of odor identity. However, for individual odors, changes in firing rate worked relatively poorly for decoding odor intensity from population activity. Analyzing odor response kinetics revealed another way odor intensity is represented in LEC. We found that increases in concentration shifted the times of peak odor responses earlier during individual sniffs and spiking became more synchronized across cells. In anterior PCx, odor representations were reported to be 'concentration invariant' when rapid population responses were studied using firing rates (*Bolding and Franks, 2018*). On the other hand, a subpopulation of PCx neurons whose latencies decreased as odor concentrations increased have been suggested to provide a temporal code for odor intensity (*Bolding and Franks, 2017*). In LEC, firing rate clearly provides some information, albeit limited, for determining odor intensity from population activity. However, temporal features of the population response can well represent intensity in LEC.

We studied LEC L2 microcircuits in more detail using voltage-clamp clamp recordings in brain slices. We found that the L2 principal neuron subtypes, fan and pyramidal cells, differed in their local and long-distance inputs. Fan cells receive contacts from more OB mitral cells than pyramidal cells. PCx inputs are also stronger onto fan cells. One simple explanation for these differences is the fact that fan cells have more extensive apical dendrites spreading through L1, where inputs converge from LOT and PCx. Intriguingly, fan cells also received markedly weaker inhibition from PV cells. This suggests that odors may produce stronger afferent excitation and weaker somatic inhibition in fan vs. pyramidal cells. Interestingly, a study using paired recordings to examine local connectivity in LEC slices (*Nilssen et al., 2018*) reported that fast spiking cell connectivity was more prevalent with fan (17 connections of 28 pairs, 61%) than pyramidal cells (3 of 11, 27%). Differences in the approach for targeting interneurons genetic targeting (ours) vs. electrophysiological properties (*Nilssen et al., 2018*) could account for the apparent discrepancy between the studies.

We studied odor-evoked responses of fan and pyramidal cells using optogenetic tagging in cre mice. If connections from OB mitral cells to LEC are distributed 'randomly' as proposed for the PCx (*Schaffer et al., 2018*; *Sosulski et al., 2011*; *Stettler and Axel, 2009*) the fact that fan cells receive more LOT input than pyramidal cells suggests that they might be more broadly tuned for odors. However, we found odor representations were actually sparser for fan compared to pyramidal cells. This observation is consistent with a recent calcium imaging study reporting that fan cells are more odor selective than pyramidal cells (*Leitner et al., 2016*). Given the smaller size of the LOT at the level of the LEC, odor representations here are presumably less influenced by direct OB input than PCx. More studies examining the distribution of OB inputs in the LEC would help understand tuning properties in this region.

We found marked differences in the timing of sniff-coupled responses in fan and pyramidal cells as odor concentration changed. Fan cells always responded earlier than pyramidal cells and their peak response times were largely concentration invariant. In contrast, the peak response times of pyramidal cells consistently shifted earlier as concentration increased. The net effect was a concentration-dependent increase in the synchrony of spiking between the two cell populations. We suspect that the stronger olfactory input and weaker PV inhibition onto fan cells, rather than broadening odor tuning, leads to their earlier activation than pyramidal cells following odor inhalation. Spike timing only carries information in relation to a reference signal. Our data suggest that fan cells could provide such a reference signal and the synchrony of fan and pyramidal cell firing could be used by downstream areas to read out odor intensity.

Like L2 LEC cells, CA1 neurons preferentially responded early during individual sniffs. Interestingly, firing rates of CA1 cells were completely insensitive to changes in odor concentration. This means that CA1 discards any firing rate changes encoding odor intensity relayed by LEC. Rather, odor intensity was encoded purely by concentration-dependent shifts in spike timing resulting in higher levels of spike synchrony as concentration increased. Temporal coding is thought to contribute to a variety of hippocampal features such as episodic memory and spatial navigation (*Dragoi, 2020*; *Eichenbaum, 2014*). The temporal coding of odor intensity in hippocampus is likely also critical for basic behaviors requiring navigation to odor sources.

## Materials and methods

All animal procedures were in accordance with protocols approved by the University of California, San Diego Institutional Animal Care and Use Committee and guidelines of the National Institutes of Health. Both female and male mice (>6 weeks old) were used for in vivo experiments. The following strains were obtained from JAX (stock number): C57BL/6J (000664), *Tbx21*[cre] (024507), *Gad2*-Cre (010802), *Pvalb*[cre] (008069), *Sst*[cre] (013044), *Calb1*[cre] (028532), Ai14(RCL-tdT) (007914), and Ai32(RCL-ChR2(H134R)/EYFP) (024109). *Ntsr1(209)*[cre] mice (Tg(Ntsr1-cre)209Gsat) were obtained from the Gensat Project. Cre[+] neurons in olfactory cortical areas have previously been characterized as pyramidal neurons (*Boyd et al., 2012*; *Stokes and Isaacson, 2010*). *NetrinG1*[cre] mice were kindly provided by Fan Wang (Duke University). Mice were maintained on a 12:12 reversed light:dark cycle and experiments were performed during the dark period.

### Viral and retrobead injections

*NetrinG1*[cre] or *Calb1*[cre] mice were anesthetized (2% isoflurane) and injected with AAV1-CAG-FLEX-tdTomato (200 nl at 20 nl/min, Addgene #28306) into the LEC (coordinates: ~3.6 mm posterior to bregma, 1.5–2.0 mm ventral to the lateral ridge on the posterior part of the temporal bone, 0.3–0.4 mm from dura at a 90° angle). Orientation of the LEC was assisted by visualizing the rhinal vein through the skull. For bead injections, *Ntsr1*[cre] × Ai14, *NetrinG1*[cre] × Ai14 mice, or *Calb1*[cre] × Ai14 were anesthetized (2% isoflurane) and injected with green retroBeads (500 nl at 100 nl/min, Lumafluor) into the DG (coordinates from bregma: 3.0 mm posterior, 2.0 mm lateral, 2.5 mm ventral from dura) or the OB (coordinates from bregma: 5.0 mm anterior, 0.8 mm lateral, 0.5 mm ventral from dura), respectively. After all injections, mice received dexamethasone (2 mg/kg) and buprenorphine (0.1 mg/kg) before return to their home cage. Mice were used for experiments 4 weeks after viral injections and 3 days after bead injections.

For in vitro recordings of PCx input to LEC, neonatal mice (postnatal day 0–2) were anesthetized by hypothermia and AAV9-hSyn-hChR2(H134R)-eYFP-WPRE-hGH was injected at four sites (23 nl/site, depths of 0.18–0.25 mm) targeting the anterior PCx based on landmarks including the superficial temporal vein and the posterior border of the eye (*Boyd et al., 2012*). Brain slices were made 3–4 weeks after injection.

### Immunohistochemistry

Mice used for histology were transcardially perfused with 4% paraformaldehyde in PBS and the brains extracted. Fixed brains were kept in 4% paraformaldehyde at 4°C for 24 hr and in 30% sucrose for 24–48 hr. Coronal sections were cut at 50 µm. Free floating sections were washed in TBS, permeabilized and blocked in TBS containing 10% horse serum and 0.2% Triton X-100, and incubated overnight with primary antibodies (mouse anti-reelin, 1:2000, MAB5364, Merck Millipore; rabbit anti-calbindin, 1:5000, CB-38a, Swant). Slices were washed in TBS, incubated for 2 hr with secondary antibodies (goat anti-mouse Alexa Fluor 488, 1:500, A11029, Thermo Fisher; goat anti-rabbit Alexa Fluor 488, 1:500, A11008, Thermo Fisher; goat anti-rabbit Alexa Fluor 647, 1:500, A32733, Thermo Fisher). Slices were mounted on slides (Vectashield medium with DAPI), imaged using a confocal microscope (Leica SP8) and subsequently analyzed with ImageJ.

### Brain slice experiments

Mice were anesthetized with isoflurane and decapitated. Brains were removed and placed into ice-cold artificial cerebrospinal fluid (aCSF) containing (in mM) 83 NaCl, 2.5 KCl$_2$, 0.5 CaCl$_2$, 3.3 MgSO$_4$, 1 NaH$_2$PO$_4$, 26.2 NaHCO$_3$, 22 glucose, and 72 sucrose, equilibrated with 95% O$_2$ and 5% CO$_2$. Coronal slices (300–400 µm thickness) containing the LEC were cut using a vibrating slicer and incubated at 35°C for 30 min. Slices were transferred to a recording chamber and superfused with aCSF containing (in mM) 119 NaCl, 5 KCl, 2.5 CaCl$_2$, 1.3 MgSO$_4$, 1 NaH$_2$PO$_4$, 26.2 NaHCO$_3$, and 22 glucose, equilibrated with 95% O$_2$ and 5% CO$_2$. All experiments were conducted at 28–30°C.

Patch-clamp recordings were performed using an upright microscope and DIC optics. Recordings were made using a Multiclamp 700 A amplifier (Molecular Devices) digitized at 20 kHz and acquired using AxographX software. For current clamp recordings, pipettes (3–6 MΩ) contained (in mM) 150 potassium gluconate, 1.5 MgCl$_2$, 5 4-(2-hydroxyethyl)-1-piperazineethanesulfonic acid (HEPES) buffer, 0.1 ethylene glycol-bis(β-aminoethyl ether)-N,N,N′,N′-tetraacetic acid (EGTA), 10 phosphocreatine,

and 2.0 Mg-ATP, pH 7.4. For voltage-clamp recordings, the internal solution contained (in mM): 130 D-gluconic acid, 130 CsOH, 5 NaCl, 10 HEPES, 0.2 EGTA, 12 phosphocreatine, 3 Mg-ATP, and 0.2 Na-GTP [pH 7.3]. Biocytin (0.2%) was added to the pipette to allow for recovery of cell morphology. Series resistance was routinely <20 MΩ and continuously monitored. Output from a collimated LED light source (470 nm, ThorLabs) was directed through the ×40 microscope objective for full-field photoactivation of ChR2.

For reconstruction of biocytin-filled cells, slices were fixed overnight in 4% PFA and then transferred into 30% sucrose. Biocytin was immunolabeled using 1:500 Alexa Fluor 488 strepavidin (Invitrogen, S32354). Slices were mounted on slides (Vectashield), imaged using a two-photon microscope (Olympus Fluoview) and subsequently analyzed with ImageJ.

## Olfactory-driven behavior

Under isoflurane anesthesia (2%), *Gad2^cre* × Ai32 or *Gad2^cre* mice were implanted with a metal head bar for head fixation. The skull above the LEC was exposed bilaterally by carefully detaching muscles from the lateral ridge of the skull. An area of bone (~750 μm diameter) over LEC was thinned and covered with transparent cyanoacrylate glue to create a window for LED illumination. This window location makes it unlikely that PCx or medial entorhinal cortex were photosuppressed and surface illumination should not impact deeper brain regions such as the ventral hippocampus. Post hoc diI marking of the window center confirmed its location over LEC. After implantation, mice received dexamethasone (2 mg/kg) and buprenorphine (0.1 mg/kg) before returning to their home cage. Behavioral experiments were conducted using Bpod State Machines (Sanworks). After >7 days of recovery, mice were water deprived (1 ml/day) and accustomed to handling and head fixation. Subsequently, mice were trained to lick for water in response to odor stimulation and trained to discriminate odor identity (isoamyl acetate 1% [vol/vol] vs. limonene 1% [vol/vol] in mineral oil) or odor intensity (ethyl tiglate 0.25% vs. 1% [vol/vol] in mineral oil) for several days in a 2AFC task until they reached the criterion of 75% correct choices. Mice indicated their choices by licking one of two lick ports for a water reward (5 μl). Respiration was recorded with a pressure sensor at the nose. Trials were initiated with a 50-ms tone. A custom-made closed-loop olfactometer was used for precisely timed and stable odor stimulation triggered by respiration. Odors were delivered for 1 s and mice were given a 2-s response time from the start of odor delivery. Error trials were followed by a 4-s timeout. On test day, mice performed the 2-AFC task with bilateral optogenetic silencing of the LEC (transcranial stimulation with fiber-coupled LEDs, Thorlabs, 470-nm, 1000-μm core fibers, 10-ms pulses, 20 Hz) randomly on 25% of trials. LED fibers were shielded to minimize light leakage. Control experiments were performed using *Gad2^cre* × Ai32 mice in which the optical windows above LEC was shielded from LED illumination or *Gad2^cre* mice that were not expressing ChR2. For analysis of response times, all trials were aligned to the onset of the first inhalation after stimulus onset.

## In vivo recordings and olfactory stimulation

Mice were implanted with a metal head bar for head fixation as described previously. The skull above the OB (coordinates from bregma: 5.0 mm anterior, 0.8 mm lateral), PCx (coordinates from bregma: 1.7 mm anterior, 2.5 mm lateral), LEC (see above and *Figure 1—figure supplement 1*), and/or CA1 (coordinates from bregma: 3.6 mm posterior, 4.4 mm lateral) was exposed and electrode insertion sites were marked on the skull. After >4 days of recovery, mice were accustomed to handling and head fixation. During recording, awake mice sat quietly in a loosely fitted plastic tube. Silicon probes were inserted into the OB (16-channel, Neuronexus, 0.5 mm from dura), PCx (32-channel, Cambridge Neurotech, 3.6 mm from dura), LEC (32-channel, Cambridge Neurotech, 0.8 mm from dura at a 90° angle), and/or CA1 (64-channel, Cambridge Neurotech, 2.2 mm from dura at a 45° angle). Electrodes were left in place for ~45 min before recordings were initiated. Signals were recorded using an Open Ephys acquisition board and digitized at 20 kHz using Open Ephys software. Probes were coated in DiI to verify recording locations post hoc. A small number of mice were used for a second recording session within several days by covering the craniotomy with silicon sealant after the first recording. Distinct probe tracks in these mice make it unlikely that the same neurons were recorded across the two sessions.

For LEC, we used 16 channels of the 32-channel linear probe (25 μm electrode spacing). The probe was inserted such that the top four channels were outside the cortex (easily determined from

the difference in noise level across sites during recording). The remaining channels (5–16) used for recording extended only 300 µm into the LEC from the cortical surface. Thus, most of the neurons we recorded should be limited to L2.

A custom-made, Arduino-controlled closed-loop olfactometer was used for precisely timed and stable odor stimulation triggered by respiration. Odor timing was verified using a fast PID (Aurora Scientific). For odor trials, charcoal-filtered air (1 l/min) was directed to glass vials containing mono-molecular odorants (*Figure 3*, 1% [vol/vol] in mineral oil) or an odor blank (mineral oil alone) using mass flow controllers. Odors were delivered in randomized order and concentrations of 0.25%, 0.33%, 0.50%, and 1% were achieved by adding 0–3 blank lines to the odor line. Recordings were made using 25 repetitions of each odor at each concentration.

For phototagging of cells in *NetrinG1*$^{cre}$ × Ai32 and *Calb1*$^{cre}$ × Ai32 mice, a fiber-coupled LED (470-nm, 400-µm core fiber, Thorlabs) was positioned within 1 mm of the exposed skull above the LEC. 5-ms light pulses were used to excite ChR2-expressing *NetrinG1*$^{cre}$ or *Calb1*$^{cre}$ neurons. Tagged units were identified based on both a significant increase in firing rate and consistent spike latency within a 5-ms window from LED onset (*Kvitsiani et al., 2013*). Given the lack of an obvious border between layers 2 and 3, the tagged pyramidal cell population we study may include a small number of L3 calbindin-expressing cells.

For recordings of responses to ChR2 activation of OB mitral cells in *Tbx21*$^{cre}$ × Ai32 mice, the fiber-coupled LED was positioned within 1 mm of the exposed skull above the OB.

### In vivo data analysis

Analysis was performed using MATLAB R2018b (MathWorks). Spikes were sorted using Kilosort2 (https://github.com/MouseLand/Kilosort; *Pachitariu, 2021*), followed by manual curation in Phy (https://github.com/cortex-lab/phy; *Rossant, 2021*) to obtain single units used for analyses. Cells were excluded from analysis if they did not maintain consistent firing and amplitude throughout the recording. For analysis of odor-evoked activity, all stimulation trials were aligned to the onset of the first inhalation of odor. Units were considered activated when mean firing rate in response to odor stimulation significantly increased compared to blank stimulation using paired *t*-tests (p < 0.05). Spike-density functions were calculated using a gaussian kernel ($\sigma$ = 10 ms) and averaged across trials. Life-time sparseness of single units was calculated based on the mean firing rate change across odors as:

$$\left( 1 - \left( \left( \sum_{j=1,N} r_j/N \right)^2 / \left( \sum_{j=1,N} r_j^2/N \right) \right) \right) / \left( 1 - 1/N \right),$$

where $r_j$ is the response of the unit to odor *j* and *N* is the total number of odors. Negative responses were set to 0 for this measure. This provides a measure of how selective the response of a unit was distributed among all odors (completely selective = 1, nonselective = 0).

PCA was performed on pseudopopulation responses of all recorded units. Euclidean distance was computed on the first three principal components in 1 ms bins. Support vector machine classification was performed on pseudopopulation responses of all recorded units in a one-vs-one coding scheme with 20% holdout validation to test the model. Statistical comparisons were performed using paired and unpaired *t*-tests, and Spearman's rank correlations.

## Acknowledgements

We are grateful to B. Nguyen for mouse training and technical support. S.H.B. is a WBP Fellow (DFG, German Research Foundation) – 445900988. H.-Z.B.Z. was a member of the UCSD-ZJUSRTP Program. This work was supported by NIH R01DC04682 and R01DC015239.

# Additional information

## Funding

| Funder | Grant reference number | Author |
|---|---|---|
| National Institute on Deafness and Other Communication Disorders | R01DC04682 | Jeffry S Isaacson |
| National Institute on Deafness and Other Communication Disorders | R01DC015239 | Jeffry S Isaacson |

The funders had no role in study design, data collection, and interpretation, or the decision to submit the work for publication.

## Author contributions

Sebastian H Bitzenhofer, Conceptualization, Data curation, Formal analysis, Funding acquisition, Investigation, Methodology, Performed and analyzed all in vivo experiments., Software, Validation, Visualization, Writing – original draft, Writing – review and editing; Elena A Westeinde, Han-Xiong Bear Zhang, Formal analysis, Investigation, Performed and analyzed in vitro experiments, Performed and analyzed in vitro experiments; Jeffry S Isaacson, Conceptualization, Data curation, Formal analysis, Funding acquisition, Investigation, Methodology, Project administration, Resources, Software, Supervision, Validation, Visualization, Writing – original draft, Writing – review and editing

## Author ORCIDs

Sebastian H Bitzenhofer (iD) http://orcid.org/0000-0003-0736-6251
Jeffry S Isaacson (iD) http://orcid.org/0000-0001-9052-5211

## Ethics

This study was performed in strict accordance with the recommendations in the Guide for the Care and Use of Laboratory Animals of the National Institutes of Health. All of the animals were handled according to approved institutional animal care and use committee (IACUC) protocols (#S977M) of the UCSD. All surgery was performed under halothane anesthesia, and every effort was made to minimize suffering.

## Decision letter and Author response

Decision letter https://doi.org/10.7554/eLife.75065.sa1
Author response https://doi.org/10.7554/eLife.75065.sa2

# Additional files

## Supplementary files

• Transparent reporting form

## Data availability

Source data is provided for each figure containing the numerical data used to generate the figures.

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
