## [Editor Report]

This work provides rigorous and high quality data regarding how neurons in the lateral entorhinal cortex (LEC) represent odor information as well as supporting a critical role of this area in odor discrimination. The LEC is an understudied area of the brain, yet critical for odor associations. This work will be of great interest to a wide neuroscience audience.

---

## [Decision Letter]

**Decision letter after peer review:**

Thank you for submitting your article "Rapid odor processing by layer 2 subcircuits in lateral entorhinal cortex" for consideration by *eLife*. Your article has been reviewed by 3 peer reviewers, including Naoshige Uchida as Reviewing Editor and Reviewer #1, and the evaluation has been overseen by Laura Colgin as the Senior Editor.

Essential revisions:

As you will see below, all the reviewers appreciate the rigor of single neuron recording and slice physiology experiments, and the importance of the results from these experiments. However, the reviewers noted some difficulties in interpreting the optogenetic experiments to demonstrate the importance of LEC in a simple odor discrimination. The reviewers discussed on this issue, and agreed that careful discussions on potential caveats of this experiment would be necessary before publication of this work in *eLife*.

1. A recent study by Kei Igrashi's group (Lee et al., Nature, 2021) showed that inhibition of fan cells, but not pyramidal neurons, impaired learning of new associations. Silencing of neither, however, affected the performance in already learned associations. The task and manipulations are somewhat different from the present study so it is difficult to compare directly but this appears to raise some concern or points to interpretational difficulty of the present results.

2. As Reviewer 2 pointed out, the animals with optogenetic silencing exhibited rapid licking relative to odor onset, indicating that the animals are displaying fast guesses before processing odor information. It remains unclear why mice exhibited fast guesses but this raises the possibility that mice reacted before having the chance to process odor information. If this possibility cannot be excluded, it remains unclear whether the impaired performance in the odor discrimination and detection tasks is due to an impairment in odor processing/discrimination or due to some other reasons that prompted mice to react before properly processing an odor.

3. There is a possibility of off-target effects. We recognize that this is true in most of previous experiments using transient manipulations. Nonetheless, given the above concerns, it would be useful to discuss potential caveats of off-target effects in the context of the present experiment.

The reviewers thought that discussing all of these issues would be important so that the results of optogenetic experiments can be placed in proper contexts. We would like to encourage the authors to include thorough discussions on the above issues as well as other issues raised below concerning the interpretation of the optogenetic experiments. To address these issues, additional experiments can be useful and we would welcome such efforts. However, we do not consider additional experiments to be essential for this revision.

*Reviewer #2 (Recommendations for the authors):*The more rapid behavioral discrimination of intensity (175 ms) vs. identity (225 ms) is interesting and reasonable re: Resulaj and Rinberg (2015 – i.e. fast, though not necessarily faster concentration discrimination), but not entirely expected from Abraham et al., (2010), Bolding and Franks (2017), or with the authors’ own results about the relative timing of identity vs. intensity information in LEC. Note that this result is not necessarily inconsistent with Bolding or with this study, because of the expectedly longer reaction vs. encoding times, but still the relative difference warrants some discussion.

The PCA trajectories are nice, and it's somewhat surprising that these returned to baseline by 200 ms. How much of the variance do the first three principal components account for?

As stated above, the slice physiology experiments are nice, are an important addition to the literature, but are descriptive and do not integrate especially meaningfully into the rest of the manuscript. Interestingly, fan cells resemble semilunar cells in PCx. Like SLs, fan cells receive stronger LOT inputs that PYRs, but unlike SLs, they also receive strong(er) intracortical inputs.

The concentration effects are small, but the concentration range is small (4-fold; c.f. 100-fold change in Bolding 2017, Roland 2017), which makes it even more remarkable that they were able to resolve their concentration-effects so clearly. This is more of an observation than a point of contention.

Figure 2, b and c – the authors should state that the red markers represent means +/- sem. The error bars are very hard to see.

The authors should discuss, even briefly, the role of the LEC in odor-cue association (e.g. Igarashi…Moser, 2014; Lee…Igarashi, 2021; etc).

Figure 3S3b – The rapid increase and decay of normalized decoding accuracy with a 2-ms window is cute but a bit contrived given how bad actual performance is (a)

Why did they need to inject flex-tdTomato into Ai14 mice?

*Reviewer #3 (Recommendations for the authors):*

Temporal coding relative to sniffing:

Since sniff-locked encoding is examined across a relatively long period of time after each inhalation (up to 300 ms), and head-fixed mice often breath much faster than 3-4 Hz, additional information would be useful to understand the interpretation of “short” vs “long” coding latencies.

A. Were any measures taken to correct for multiple sniffs being included in the “long” latency period? From Supp. Figure 1b it does appear that some trials would include a second sniff happening in the time period between 200-300 after the first sniff onset. This could contribute to noise in the late response that would not be present in the early response.

B. Would the short vs. long latency coding results look different if only sniff cycles without another sniff in the long latency bins were included in the analysis? What about separately analyzing “short” and “long” sniff cycles (similar to Shusterman, et al., 2018)?

C. Could the authors comment on why <100 ms was chosen as an “early” spiking response relative to inhalation? It seems that what is considered early and late in the sniff cycle can vary across studies, so a rationale here would be useful.

Optogenetic block of LEC:

The advantage of optogenetics in this context is the precise control that the researchers have over when the disruption is applied to the LEC. The authors argue that early neural activity in the first sniff following odor application is essential for the animals’ performance on the task. They could directly test this by only blocking the LEC after this early phase. The prediction would be that on trials with “late” LEC block animals would have much improved performance compared to those that experience complete block. This technique has been used previously in the olfactory system (the “optogenetic masking” technique of Wilson, et al., 2017 is similar) to investigate “primacy coding” in the olfactory bulb and would greatly enhance the impact of the current work.

[Editors’ note: further revisions were suggested prior to acceptance, as described below.]

Thank you for resubmitting your work entitled “Rapid odor processing by layer 2 subcircuits in lateral entorhinal cortex” for further consideration by *eLife*. Your revised article has been evaluated by Laura Colgin (Senior Editor) and a Reviewing Editor.

The manuscript has been improved but there are some remaining issues that need to be addressed, as outlined below:

The authors responded to the reviewers’ previous concerns mostly adequately. However, we have one request before accepting this manuscript.

Some issues such as the impact of fast guesses in performance appear to remain as an issue of further discussion, but the current manuscript provides the data for the readers to evaluate. We still would like to request one change regarding the analysis of sequential activity. As the authors’ new analysis in the rebuttal shows, the way the data is currently presented in the manuscript overestimates the extent of sequential activity. The authors’ new analysis using cross validation (sorting with the odd trials and plotting only the even trials) shows a lesser extent of sequence, with the LEC almost losing the sequential pattern. Although there are still some papers that use the authors’ original method (sorting with all trials and plotting all trials) in the field, this practice must be corrected. Therefore, we request that all of the sequential activity figures be replaced to the cross-validated ones (e.g. sorting by the odd trials, and plotting the even trials). Given the apparent loss of sequence in LEC, it would be useful if the authors provide some quantitative analysis of sequence.

---

## [Author Response]

1. A recent study by Kei Igrashi’s group (Lee et al., Nature, 2021) showed that inhibition of fan cells, but not pyramidal neurons, impaired learning of new associations. Silencing of neither, however, affected the performance in already learned associations. The task and manipulations are somewhat different from the present study so it is difficult to compare directly but this appears to raise some concern or points to interpretational difficulty of the present results.

Actually, this paper does not raise concerns and actually supports our findings. It is correct that Lee et al., 2021 show that suppressing activity in either fan or pyramidal cells individually did not disrupt performance of learned odor associations. However, Extended Figure 8 of that study also shows that performance of learned odor associations drops to chance during LEC infusion of the GABAA-receptor agonist muscimol. Thus, they find that a manipulation that suppresses activity in fan and pyramidal cells simultaneously completely disrupts odor discrimination. This is entirely consistent with our observation using optogenetic silencing and that is why we cited Lee et al., 2021 paper in the original Discussion.

2. As Reviewer 2 pointed out, the animals with optogenetic silencing exhibited rapid licking relative to odor onset, indicating that the animals are displaying fast guesses before processing odor information. It remains unclear why mice exhibited fast guesses but this raises the possibility that mice reacted before having the chance to process odor information. If this possibility cannot be excluded, it remains unclear whether the impaired performance in the odor discrimination and detection tasks is due to an impairment in odor processing/discrimination or due to some other reasons that prompted mice to react before properly processing an odor.

As discussed below, although a 50 ms auditory cue immediately preceded odor delivery, lick timing was aligned to the first odor inhalation (“time 0”) and we excluded the small fraction of trials in which licks preceded inhalation. We reanalyzed the excluded trials and found no difference in the fraction of trials with licks before the first odor inhalation with the LED on vs. LED off. Thus, our results are inconsistent with the idea that mice make rapid guesses (“ballistic licks”) triggered by the tone cue during LEC silencing.

3. There is a possibility of off-target effects. We recognize that this is true in most of previous experiments using transient manipulations. Nonetheless, given the above concerns, it would be useful to discuss potential caveats of off-target effects in the context of the present experiment.

It is unclear what off-target effects are raised by this point, but our use of transcranial illumination through an optical window over LEC makes it unlikely to us that we are inactivating deep structures or those removed from this cortical area like piriform cortex. Nonetheless, we have added to the Discussion caveats related to the interpretation of behavioral results based on acute optogenetic silencing:

“While acute optogenetic silencing of LEC disrupted performance in behavioral tasks requiring discrimination of odor identity and intensity, our findings do not necessarily prove that odor discrimination within the LEC is required for behavior. It could be the case that LEC is essential for the retrieval of learned odor-reward associations while discrimination is provided elsewhere (i.e. piriform cortex). Nonetheless, our results are consistent with LEC playing a critical role in odor-driven behavior. An additional caveat to experiments using optogenetic silencing, is that acute perturbations could disrupt behavior by removing “permissive” activity for downstream structures, rather than implicating the target circuit as “instructive” for the behavior (Wolff and Ölveczky, 2018). Future experiments examining the effects of chronic LEC inactivation on odor discrimination will be useful to probe this possibility.”

Reviewer #2 (Recommendations for the authors):The more rapid behavioral discrimination of intensity (175 ms) vs. identity (225 ms) is interesting and reasonable re: Resulaj and Rinberg (2015 – i.e. fast, though not necessarily faster concentration discrimination), but not entirely expected from Abraham et al., (2010), Bolding and Franks (2017), or with the authors' own results about the relative timing of identity vs. intensity information in LEC. Note that this result is not necessarily inconsistent with Bolding or with this study, because of the expectedly longer reaction vs. encoding times, but still the relative difference warrants some discussion.

While this distinction in d’ is potentially interesting, we did not see any obvious difference when comparing the time courses of correct choices over time (data added to Figure 2-supplement 1). We prefer to err on the side of caution and not make a claim about the timing of behavior discriminating identity vs. intensity.

The PCA trajectories are nice, and it's somewhat surprising that these returned to baseline by 200 ms. How much of the variance do the first three principal components account for?

As stated in the legend for Figure 3f, the three principal components account for 69% of the variance.

As stated above, the slice physiology experiments are nice, are an important addition to the literature, but are descriptive and do not integrate especially meaningfully into the rest of the manuscript. Interestingly, fan cells resemble semilunar cells in PCx. Like SLs, fan cells receive stronger LOT inputs that PYRs, but unlike SLs, they also receive strong(er) intracortical inputs.

As stated in the Discussion, we think the slice experiments raise the possibility that fan cells respond earlier to odors than pyramidal cells due to the fact that they receive more olfactory inputs.

The concentration effects are small, but the concentration range is small (4-fold; c.f. 100-fold change in Bolding 2017, Roland 2017), which makes it even more remarkable that they were able to resolve their concentration-effects so clearly. This is more of an observation than a point of contention.Figure 2, b and c – the authors should state that the red markers represent means +/- sem. The error bars are very hard to see.

Added, “Red circles, mean ± SEM.”

The authors should discuss, even briefly, the role of the LEC in odor-cue association (e.g. Igarashi…Moser, 2014; Lee…Igarashi, 2021; etc).

We included this in our expanded points regarding behavior in the Discussion:

“Previous behavioral studies have differed regarding the importance of LEC for odor discrimination. […] Future experiments examining the effects of chronic LEC inactivation on odor discrimination will be useful to probe this possibility.”

Figure 3S3b – The rapid increase and decay of normalized decoding accuracy with a 2-ms window is cute but a bit contrived given how bad actual performance is (a)

It is true that performance in that 2 ms window is poor, but it does show that information over short timescales is more transient and fits with the PCA results.

Why did they need to inject flex-tdTomato into Ai14 mice?

Thanks for spotting that error in the Methods! Flex-tdTomato was injected into NetrinG1- and Calbindin-cre mice that were not crossed with the Ai14 line.

Reviewer #3 (Recommendations for the authors):Temporal coding relative to sniffing:Since sniff-locked encoding is examined across a relatively long period of time after each inhalation (up to 300 ms), and head-fixed mice often breath much faster than 3-4 Hz, additional information would be useful to understand the interpretation of "short" vs "long" coding latencies.A. Were any measures taken to correct for multiple sniffs being included in the "long" latency period? From Supp. Figure 1b it does appear that some trials would include a second sniff happening in the time period between 200-300 after the first sniff onset. This could contribute to noise in the late response that would not be present in the early response.

In our hands, mice respond with rapid sniffing when they are not acclimatized to head-fixation and also sniff in response to the first few odor applications of a recording session. We always habituated mice to head-fixation and applied odors several times before recording to reduce rapid sniffing. In addition, we analyzed the timing of the 2^nd^ inhalation of odor in all of our recordings from Figure 1 (n = 6 mice) and found very few trials when a second sniff occurred within 300 ms of the first inhalation (Author response image 1).

**Author response image 1. sa2fig1:** 

B. Would the short vs. long latency coding results look different if only sniff cycles without another sniff in the long latency bins were included in the analysis? What about separately analyzing "short" and "long" sniff cycles (similar to Shusterman, et al., 2018)?

We did not explicitly include an analysis of the difference between “short” (within 100 ms of inhalation) and “long” (200-300 ms after inhalation). Our PCA results indicate that there is not much information added after 200 ms.

C. Could the authors comment on why <100 ms was chosen as an "early" spiking response relative to inhalation? It seems that what is considered early and late in the sniff cycle can vary across studies, so a rationale here would be useful.

As we stated in the Results:

“Across all cells (n=576), odor-evoked firing peaked in a time window 50-100 ms from the start of odor inhalation, considerably earlier than respiration coupled activity observed in the absence of applied odor (blank response, Figure 3b). We thus used this 50 ms time window for analysis of rapid odor evoked activity in LEC.”

Optogenetic block of LEC:The advantage of optogenetics in this context is the precise control that the researchers have over when the disruption is applied to the LEC. The authors argue that early neural activity in the first sniff following odor application is essential for the animals' performance on the task. They could directly test this by only blocking the LEC after this early phase. The prediction would be that on trials with "late" LEC block animals would have much improved performance compared to those that experience complete block. This technique has been used previously in the olfactory system (the "optogenetic masking" technique of Wilson, et al., 2017 is similar) to investigate "primacy coding" in the olfactory bulb and would greatly enhance the impact of the current work.

This is a great idea and one we also considered, but it would require very precise respiration-triggered optogenetic stimulation and very high number of LED trials. This would be great to test in future studies.

[Editors’ note: further revisions were suggested prior to acceptance, as described below.]

The manuscript has been improved but there are some remaining issues that need to be addressed, as outlined below:The authors responded to the reviewers' previous concerns mostly adequately. However, we have one request before accepting this manuscript.Some issues such as the impact of fast guesses in performance appear to remain as an issue of further discussion, but the current manuscript provides the data for the readers to evaluate. We still would like to request one change regarding the analysis of sequential activity. As the authors' new analysis in the rebuttal shows, the way the data is currently presented in the manuscript overestimates the extent of sequential activity. The authors' new analysis using cross validation (sorting with the odd trials and plotting only the even trials) shows a lesser extent of sequence, with the LEC almost losing the sequential pattern. Although there are still some papers that use the authors' original method (sorting with all trials and plotting all trials) in the field, this practice must be corrected. Therefore, we request that all of the sequential activity figures be replaced to the cross-validated ones (e.g. sorting by the odd trials, and plotting the even trials). Given the apparent loss of sequence in LEC, it would be useful if the authors provide some quantitative analysis of sequence.

This second revision contains panels in Figures 1, 6 and 7 substituted with plots requested by the editor (cross-validation of sorted PSTHs using even trials plotted relative to sorted odd trials).